# Mexican Biobank advances population and medical genomics of diverse ancestries

Mashaal Sohail[1,2,16 ✉], María J. Palma-Martínez[1,19], Amanda Y. Chong[3,19], Consuelo D. Quinto-Cortés[1,19], Carmina Barberena-Jonas[1], Santiago G. Medina-Muñoz[1], Aaron Ragsdale[1,17], Guadalupe Delgado-Sánchez[4], Luis Pablo Cruz-Hervert[4,5], Leticia Ferreyra-Reyes[4], Elizabeth Ferreira-Guerrero[4], Norma Mongua-Rodríguez[4], Sergio Canizales-Quintero[4], Andrés Jimenez-Kaufmann[1], Hortensia Moreno-Macías[6,7], Carlos A. Aguilar-Salinas[8], Kathryn Auckland[3], Adrián Cortés[9], Víctor Acuña-Alonzo[10], Christopher R. Gignoux[11], Genevieve L. Wojcik[12], Alexander G. Ioannidis[13], Selene L. Fernández-Valverde[1,18], Adrian V. S. Hill[3,14], María Teresa Tusié-Luna[6], Alexander J. Mentzer[3,9 ✉], John Novembre[2,15], Lourdes García-García[4,20 ✉] & Andrés Moreno-Estrada[1,20 ✉]

Latin America continues to be severely underrepresented in genomics research, and fine-scale genetic histories and complex trait architectures remain hidden owing to insufficient data[1]. To fill this gap, the Mexican Biobank project genotyped 6,057 individuals from 898 rural and urban localities across all 32 states in Mexico at a resolution of 1.8 million genome-wide markers with linked complex trait and disease information creating a valuable nationwide genotype–phenotype database. Here, using ancestry deconvolution and inference of identity-by-descent segments, we inferred ancestral population sizes across Mesoamerican regions over time, unravelling Indigenous, colonial and postcolonial demographic dynamics[2–6]. We observed variation in runs of homozygosity among genomic regions with different ancestries reflecting distinct demographic histories and, in turn, different distributions of rare deleterious variants. We conducted genome-wide association studies (GWAS) for 22 complex traits and found that several traits are better predicted using the Mexican Biobank GWAS compared to the UK Biobank GWAS[7,8]. We identified genetic and environmental factors associating with trait variation, such as the length of the genome in runs of homozygosity as a predictor for body mass index, triglycerides, glucose and height. This study provides insights into the genetic histories of individuals in Mexico and dissects their complex trait architectures, both crucial for making precision and preventive medicine initiatives accessible worldwide.

The architecture of complex traits in humans can be fully understood only in the context of history. Present-day Mexico covers seven cultural regions, including much of Mesoamerica, with rich civilizational histories[9]. Archaeological and anthropological approaches have been used to regionalize Mexico into the north of Mexico, the north of Mesoamerica, the centre, occident and Gulf of Mexico, Oaxaca (referring here to the Oaxaca cultural region) and the Mayan region[10] (Fig. 1a). These regions are based on specific Indigenous civilizations and cultures, which flourished early in the Mayan region, Oaxaca, and the occident and the Gulf of Mexico, and later in the centre and north of Mesoamerica. Such histories have also been used to classify Mesoamerican chronology into preclassical, classical, postclassical, colonial and postcolonial periods[11].

In the past 500 years, Spanish colonization has left an indelible mark on this Indigenous tapestry. In a colonial and postcolonial context, genetic ancestries that trace principally to Western Europe, West Africa

[1]Unidad de Genómica Avanzada (UGA-LANGEBIO), Centro de Investigación y Estudios Avanzados del IPN (Cinvestav), Irapuato, Mexico. [2]Department of Human Genetics, University of Chicago, Chicago, IL, USA. [3]The Wellcome Centre for Human Genetics, University of Oxford, Oxford, UK. [4]Instituto Nacional de Salud Pública (INSP), Cuernavaca, Mexico. [5]División de Estudios de Posgrado e Investigación, Facultad de Odontología, Universidad Nacional Autónoma de México (UNAM), Mexico City, Mexico. [6]Unidad de Biología Molecular y Medicina Genómica, Instituto de Investigaciones Biomédicas UNAM/Instituto Nacional de Ciencias Médicas y Nutrición Salvador Zubirán, Mexico City, Mexico. [7]Universidad Autónoma Metropolitana, Mexico City, Mexico. [8]Division de Nutrición, Instituto Nacional de Ciencias Médicas y Nutrición Salvador Zubirán, Mexico City, Mexico. [9]Big Data Institute, Li Ka Shing Centre for Health Information and Discovery, University of Oxford, Oxford, UK. [10]Escuela Nacional de Antropología e Historia (ENAH), Mexico City, Mexico. [11]Colorado Center for Personalized Medicine, University of Colorado Anschutz Medical Campus, Aurora, CO, USA. [12]Department of Epidemiology, Johns Hopkins Bloomberg School of Public Health, Baltimore, MD, USA. [13]Department of Biomedical Data Science, Stanford University, Stanford, CA, USA. [14]The Jenner Institute, University of Oxford, Oxford, UK. [15]Department of Ecology and Evolution, University of Chicago, Chicago, IL, USA. [16]Present address: Centro de Ciencias Genómicas (CCG), Universidad Nacional Autónoma de México (UNAM), Cuernavaca, Mexico. [17]Present address: Department of Integrative Biology, University of Wisconsin-Madison, Madison, WI, USA. [18]Present address: School of Biotechnology and Biomolecular Sciences and the RNA Institute, The University of New South Wales, Sydney, New South Wales, Australia. [19]These authors contributed equally: María J. Palma-Martínez, Amanda Y. Chong, Consuelo D. Quinto-Cortés. [20]These authors jointly supervised this work: Lourdes García-García and Andrés Moreno Estrada. ✉e-mail: mashaal@ccg.unam.mx; alexander.mentzer@ndm.ox.ac.uk; garcigar@insp.mx; andres.moreno@cinvestav.mx

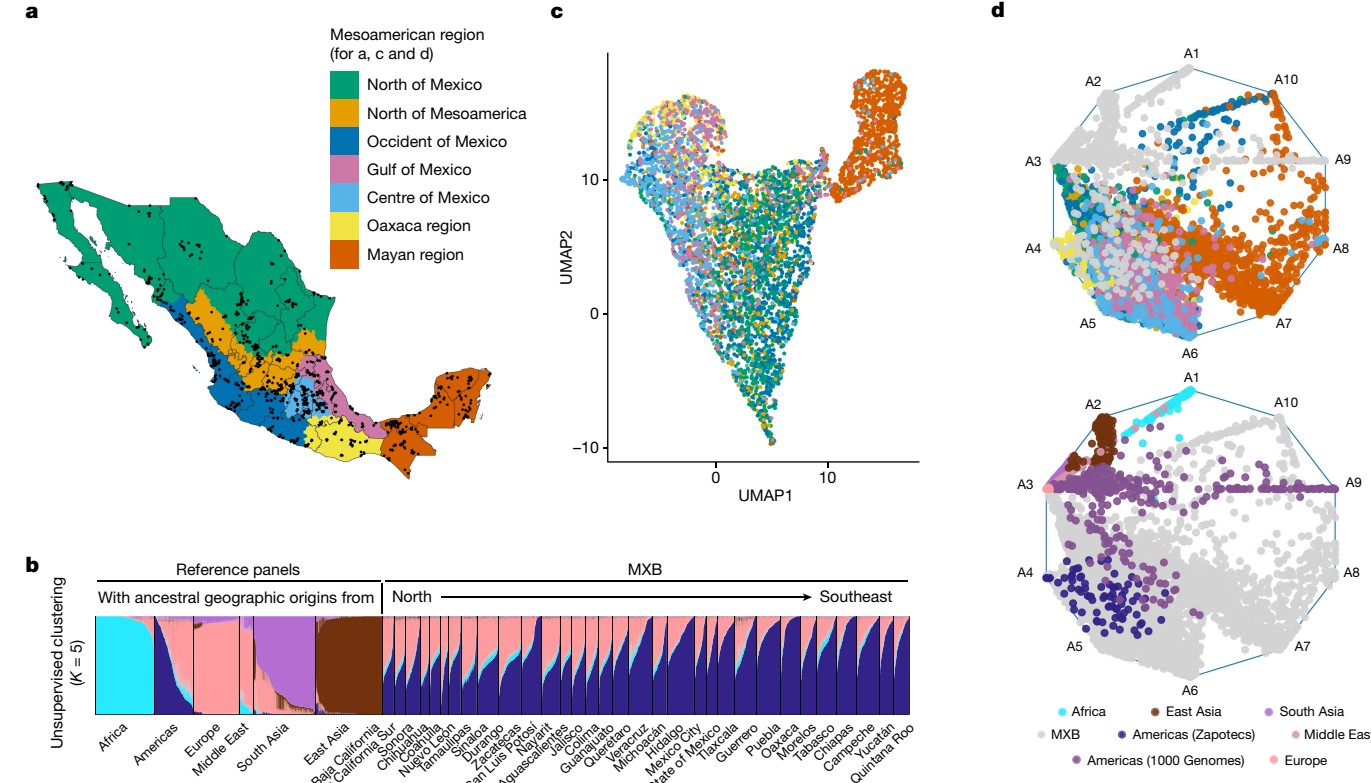

**Fig. 1 | Mosaic ancestral patterns in the MXB and the genetic diversity within Mexico. a**, Sampling for the MXB ($n$ = 5,812 individuals with latitude and longitude values), showing Mexico regionalized into Mesoamerican regions according to an anthropological and archaeological context. **b**, Unsupervised clustering using ADMIXTURE and global reference panels ($n$ = 9,007 including MXB) from the 1000 Genomes Project, the Human Genome Diversity Project and the Population Architecture using Genomics and Epidemiology Study. **c**, Uniform manifold approximation and projection (UMAP) analysis of MXB ($n$ = 5,622) coloured by Mesoamerican region. **d**, Archetypal analysis of MXB ($n$ = 5,833) with reference global individuals as in **b**, coloured by region (top) or in grey (bottom). This approach determines each individual's position in a ten-dimensional space that in this visualization is reduced to two dimensions. Reference individuals (bottom) are coloured using ADMIXTURE inferred clusters from **b**. For example, for the Americas (1000 Genomes) and Middle East, where multiple clusters are inferred, a colour combining these cluster colours is used.

and East Asia can be identified in present-day Mexicans[12–16]. These genetic ancestries vary in structure and timing between Mesoamerican regions and give rise to extensive fine-scale population substructure and ancestry sources across Mexico[12–16]. Further, such varying genetic histories, as captured by ancestry distributions, have been shown to affect variation in complex traits such as lung force capacity[12], and a number of other complex traits and diseases[17].

Nevertheless, a large gap remains in the representation of Mexicans from across Mexico in cohorts with linked genotypes and phenotypes. Such representation could enable finer-scale studies of genetic histories and a better understanding of complex trait architectures among individuals with diverse ancestries from the Americas and those living in rural areas[18]. Past analyses on complex traits have been limited to studying individuals from the USA and Mexico City[12,17]. They have also not simultaneously modelled the influence on complex trait variation of a rich array of genetic and environmental factors as is possible with a nationwide biobank.

To bridge this gap, we launched the Mexican Biobank (MXB) project, densely genotyping 6,057 individuals from 898 localities distributed nationwide (Supplementary Figs. 1 and 2) recruited by the National Institute of Public Health (Instituto Nacional de Salud Pública) across all 32 states of Mexico. To select the samples for genomic and biochemical characterization, we enriched for those individuals that speak an Indigenous language while maximizing geographic coverage and the inclusion of rural localities (about 70% of the MXB; Supplementary Figs. 2–5). Of the participants in the MXB, 70% are female, and it comprises data for individuals born between 1910 and 1980

(Supplementary Table 1) who were genotyped at about 1.8 million single nucleotide polymorphisms (SNPs) and have linked information for complex traits, sociocultural and biogeographical markers (Supplementary Table 2).

Here, we leverage rich archaeological and anthropological information to guide a regionalized analysis of Mexico, and harness the power of genome-wide local ancestry estimation and identity-by-descent (IBD) segments to decipher fine-scale genetic histories using ancestry-specific approaches to denote origins and historical population size changes[4,19]. We reveal a very heterogeneous landscape of both, painting a genetically informed picture of varying demographic trajectories in Mesoamerican regions, including colonial migrations and dynamics. We further investigate the role of these evolutionary histories as captured by proxies of genetic ancestries in shaping genetic variation and complex trait patterns in Mexico today. We show that these histories result in marked geographic and ancestry-specific patterns in the distributions of runs of homozygosity (ROH) and of the genomic burden of rare deleterious variants. We carry out GWAS analyses across 22 binary and quantitative traits, and compare the prediction performance of polygenic scores computed using our GWAS or UK Biobank (UKB) GWAS data. Last, given that evolutionary histories (captured by genetic ancestries) could associate specific trait-relevant genotypes with certain genetic backgrounds, we study the impact of genetic ancestries, portions of the genome in ROH, polygenic scores and other sociocultural and biogeographical factors on creating variation in complex and medically relevant traits in Mexico.

## Diverse ancestries across timescales

We begin by analysing the population structure in the MXB at different geographic resolutions and timescales (see the section entitled 'Note on genetic ancestries' in the Methods; Fig. 1 and Supplementary Figs. 6–24). Given the history of Mexico, in which genetic lineages are expected to trace back to disparate geographic regions (for example, the Americas, Western Europe, West Africa and East Asia) in the past approximately 500 years, we first analyse each individual in a framework that infers proportions of genetic ancestries on the basis of genetic similarity to other individuals (using ADMIXTURE[20]) in a global reference sample. We use a similar approach to label local segments across the genomes of the study individuals. We use the term 'ancestries from the Americas' when referring to genetic ancestries that derive from genetic ancestors living in the Americas before European colonization; these have also been referred to as Indigenous ancestries, and in some places below we also use this term (Fig. 1b, Supplementary Figs. 11 and 12 and Supplementary Table 3).

Higher proportional ancestries from the Americas are inferred in Mexico's central and southern states, compared to the northern states, and ancestries from West Africa are observed in every state[21] (Supplementary Table 3) in agreement with historical records of shipping voyages from the transatlantic slave trade[21,22] (Supplementary Fig. 14). We note the presence of a small but substantial proportion of ancestries from East Asia in almost every state (0–2.3%), the highest in the state of Guerrero (2.3%), and an even more modest proportion of ancestries from South Asia in most states as well (0–0.8%). These probably reflect migrations from Asia to Mexico dating to the Manila Galleon trade in the sixteenth and seventeenth centuries[16,23–26], and later nineteenth- and twentieth-century migrations from China and Japan, especially to the north of Mexico[27–29].

We observe the most significant genetic differentiation along a north-to-southeast cline in Mexico (measured using $F_{ST}$, which is an index quantifying the proportion of the total genetic variance contained in subpopulations (S) relative to the total genetic variance (T); Supplementary Figs. 15–18). When considering autosomes of only individuals with ≥90% proportion of ancestries from the Americas (inferred using ADMIXTURE), the Mayan region of Chiapas, Tabasco, Yucatan, Quintana Roo and Campeche show relatively larger $F_{ST}$ values with the other regions (Supplementary Figs. 17 and 18). This distinction is also apparent using ADMIXTURE-inferred ancestral clusters (Supplementary Fig. 13) and dimensionality reduction techniques highlighting this population substructure within Mexico (Fig. 1c,d and Supplementary Figs. 7–20). Individuals from the Mayan region tend to cluster mostly together, but overlap with individuals from the Gulf of Mexico and central Mexico, consistent with oral histories. In the rest of the regions, subtle substructure mirroring Mesoamerican geography is visible in the MXB, probably reflecting both unique local demographic histories of Indigenous ancestries and the effects of movement and mating among the different regions. Compared to previous sampling and analyses that focused on Indigenous groups with varying degrees of isolation in Mexico[12], the MXB reveals lower average levels of $F_{ST}$ and substructure, probably owing to the broader sampling (although the substructure presented by the Mayan region is more apparent in the MXB). The method of ref. 5 further highlights the ancestral diversity reflected by the MXB samples that are represented as mixtures of multiple sources (Supplementary Fig. 22) in the presence of global references (Fig. 1d and Supplementary Figs. 21–23). Individuals from the same region (for example, the Mayan region) are modelled as mixtures of several sources, reflecting the diversity of ancestry variation within this and other Mesoamerican regions. Given this variation among ancestries from the Americas and the unique power given by the MXB to explore its impact on complex trait variation, we also obtain an axis of variation within ancestries from the Americas (Supplementary Fig. 24 and Supplementary Table 4).

## Genetic histories inferred within Mexico

Contemporary Mexicans derive ancestries predominantly from diverse lineages found in the Americas, Western Europe and West Africa. These ancestral sources have different demographic histories before their arrival in present-day Mexico and probably after their arrival within different Mesoamerican regions. To reveal the history of effective population sizes ($N_e$) of these three ancestries in the MXB, we analyse IBD segments[4,30] stratified by local ancestry inference for each Mesoamerican region[4] (Fig. 2).

We observe fine-scale variation in $N_e$ trajectories for Indigenous lineages which we interpret in the context of the different cultural histories of Mesoamerican regions[9] (Fig. 2). As generational time can vary, we present our analysis at two extremes of 20 and 30 years per generation[31] (Supplementary Figs. 25 and 2c, respectively). Chronologically speaking, archaeologists document that Mesoamerican civilizations flourished first in the Mayan region, in Oaxaca, in the occident and in the Gulf of Mexico. In these regions, we observe large $N_e$ already in the classical period (250–900 CE)[32]. For example, in the Gulf, where we observe high $N_e$ since the preclassical period (2500 BCE–250 CE), there is archaeological evidence, among a myriad of other groups, of the Olmecs in the preclassical period, the Totonacs in the classical period and the Huastecs in the postclassical period (900–1521 CE)[33]. In Oaxaca, we observe $N_e$ rapidly growing in the preclassical to the classical period, in line with archaeological inferences that the Zapotecs were already starting to create sedentary settlements in the preclassical period followed by a rise in social and political structures in the classical period. The subsequent postclassical period was characterized by militarism and warfare[34], and our genetic evidence suggests a population decline towards the end of the postclassical period. In the Yucatan peninsula, the Maya had a prominent civilizational spread in the classical period (peak $N_e$ observed). They started going through a slow decline only in the postclassical period due to what archaeologists have inferred as a combination of different political and ecological factors, and this trajectory is supported in the $N_e$ trend[32].

These patterns contrast with those of the centre and north of Mesoamerica, where the Aztec empire had a stronghold most recently; there we see increasing $N_e$ in the postclassical right before the arrival of the Spaniards and into part of the colonial period, after which we start to see a population decline in $N_e$. The decrease in $N_e$ after the arrival of the Spaniards is most prominent in the centre and north of Mesoamerica. In Oaxaca and the Mayan region, where Indigenous ancestries from the Americas are most prevalent today as evidenced by the ADMIXTURE analysis (Supplementary Table 3), the decrease in $N_e$ is followed by an increase in the postcolonial period.

Concurrently, we observe that ancestries from Western Europe that entered the contemporary Mexican gene pool went through a sharp decline in effective population size during the colonial period. The extent of the founder effect varied by region, with the strongest effect seen in Oaxaca and the Mayan region (Supplementary Figs. 26 and 27). Ancestries from West Africa in Mexico revealed stronger founder effects that varied by region, with $N_e$ ranging between $10^3$ and $10^4$ in the colonial period. The population size in the postcolonial period continued to grow in some regions such as the occident and north of Mexico and the Mayan region, compared to others (Supplementary Figs. 28 and 29). Consistent with previous results on self-identified Indigenous groups[13,14], our results on the MXB individuals highlight the heterogeneity of group histories across the Mesoamerican regions as well as the expansion of Indigenous lineages in the postcolonial period in several regions.

We further generated 'admixture graphs'[6] for individuals from the Mesoamerican regions to investigate their shared history by using an ancestry-specific approach and limiting the analysis to genomic segments with ancestries from the Americas. The admixture graph approach models the different Mesoamerican regions as populations

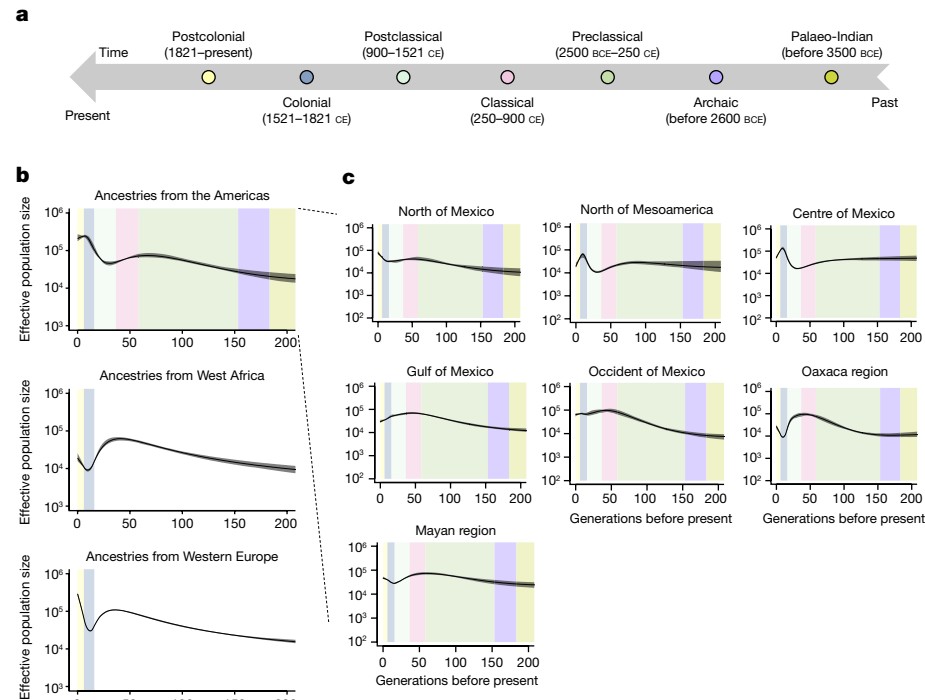

**Fig. 2 | Effective population size ($N_e$) values across ancestries and geographies reveal the histories present within Mexico. a**, Mesoamerican chronology colouring different periods in Mesoamerican history using an anthropological and archaeological context. **b**, Ancestry-specific effective population size ($N_e$) changes over the past 200 generations across Mexico ($n = 5,436$) inferred using IBD tracts, coloured by chronology from **a** assuming 30 years per generation.

**c**, Ancestry-specific effective population size ($N_e$) changes over time for ancestries from the Americas in different regions of Mexico (see Supplementary Figs. 25–29 for other generation intervals and ancestries). $n = 1,177, 640, 952, 590, 820, 315$ and $938$ for the north of Mexico, north of Mesoamerica, centre of Mexico, Gulf of Mexico, occident of Mexico, Oaxaca region and the Mayan region, respectively.

in a progression of splits (Extended Data Fig. 1a and Supplementary Fig. 30), providing information about the genetic relationships among the different regions. We can observe a clear progression of splits among the populations from north to south, with the north of Mexico splitting first, followed by the common ancestor of the north of Mesoamerica and the occident of Mexico, followed by the common ancestor of the remaining regions. Notably, the centre of Mexico and the Mayan region are related, consistent with previous suggestions based on IBD[12] and our population structure results, and both share a common ancestral source with Oaxaca and the Gulf of Mexico. These results further strengthen evidence for an Atlantic coastal corridor of gene flow between the Yucatan peninsula and central Mexico and the Gulf of Mexico previously posited in ref. 12. As demographic histories can affect patterns of genetic variation, such as distributions of ROH and of the genomic burden of deleterious variants, we next evaluate these metrics.

### Impact of genetic histories on variation

We analyse the patterns of ROH in the MXB including how they vary across geography and genetic ancestry proxies (inferred from ADMIXTURE). ROH patterns help further illuminate the demographic and mating histories of Mexicans[35], and are especially relevant for variation in complex traits when trait-relevant variation is affected by partially recessively acting alleles[36]. We identify ROH (≥1 Mb) in the MXB and observe that both the number of ROHs and the total length of ROH per individual increase as we move from north to southeast in the country (Supplementary Fig. 31). We confirm that this is primarily due to individuals with a higher inferred proportion of genetic ancestries from the Americas also having more ROH, particularly small ROH (smaller than those expected from recent consanguinity; for example, <8 Mb), in their genomes (Fig. 3a, Supplementary Figs. 32 and 33 and Supplementary Table 5). The appearance of many small ROHs

indicates coalescences occurring at a period in the more distant past; for example, due to an ancient bottleneck or relatively small historical population size[37].

Further, we observe that ROH found on Indigenous genomic segments are more frequent in younger individuals compared to older individuals (Spearman's $\rho = 0.31$, $P = 0.016$; Fig. 3b). We corroborate that this correlation with birth year primarily derives from small ROH ($\rho = 0.35$, $P = 0.006$), and small ROH found on Indigenous genomic segments ($\rho = 0.39$, $P = 0.002$; Fig. 3b). The result is at least partly due to younger individuals having higher proportions of Indigenous ancestries compared to older individuals, especially in the rural localities (Supplementary Figs. 34 and 35), and agrees with recent observations about ancestry and ROH made in Mexican Americans[17]. We also confirm that this observation is not due to sampling bias (see the sampling ascertainment note in the Supplementary Information). The observation of higher ancestries from the Americas in younger individuals in rural areas may be due to higher fertility rates in rural areas or individuals with other ancestries moving out from rural to urban areas.

We also investigate the effects of demographic histories on the frequency distribution of genetic variants. This analysis is motivated by previous theoretical and empirical work showing that undergoing a bottleneck changes the allele frequency distribution in the group that experienced the bottleneck[38–40], while leaving the overall sum of deleterious alleles per individual ('deleterious mutational burden') unchanged[39,41,42]. In particular, rare variants are lost or increase in frequency after the bottleneck.

We evaluate this effect by computing the genome-wide sum of intergenic, synonymous and putatively deleterious (predicted-damaging missense and loss of function) alleles per individual. When considering only rare alleles (derived allele frequency ≤ 5%), we observe that individuals with higher ancestry proportions from the Americas carry fewer rare derived alleles across variant types (strongest effect

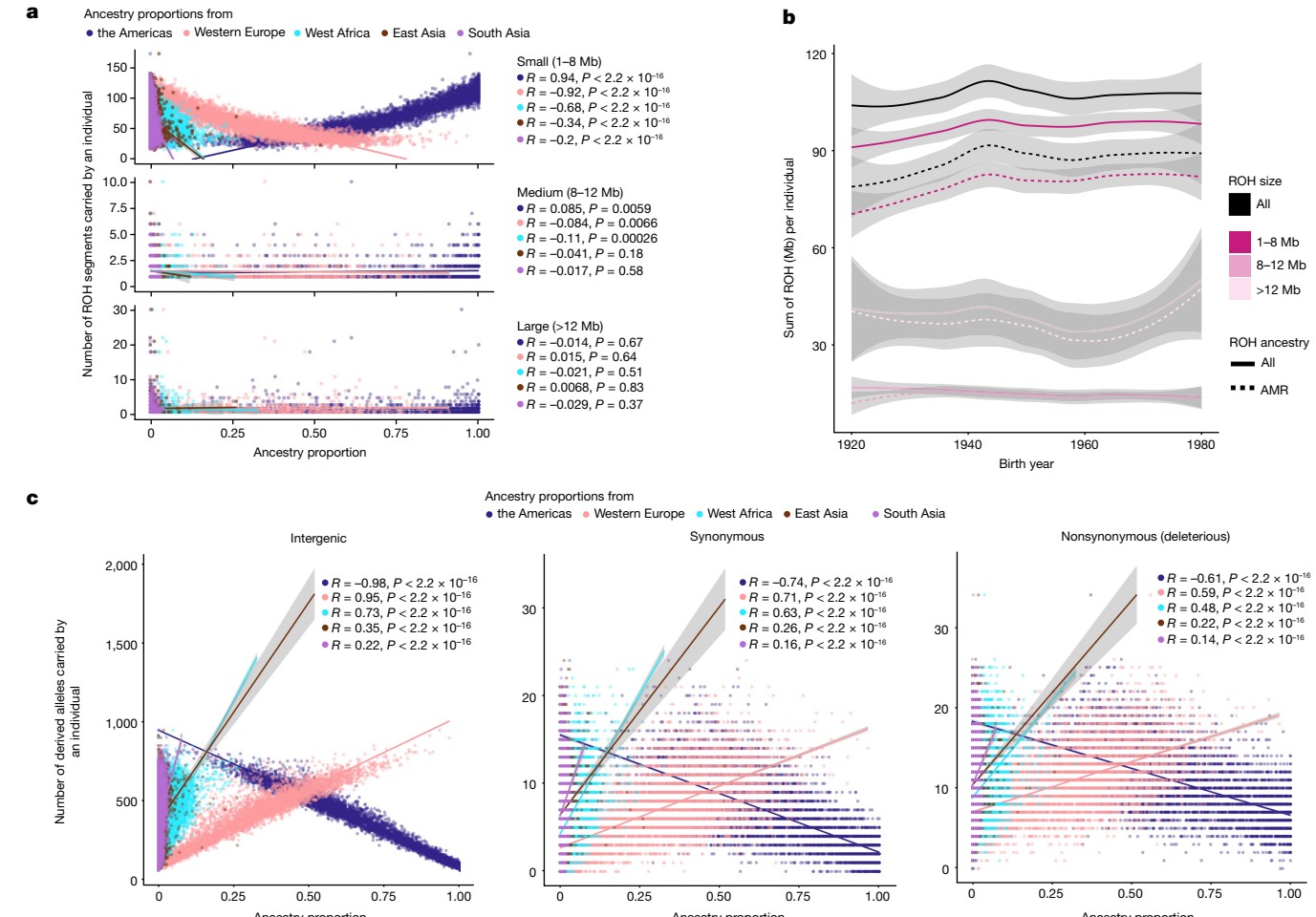

**Fig. 3 | Demographic histories affect patterns of genetic variation in Mexico.**
**a**, Small ROH prevalence is correlated with ancestry proxies inferred from ADMIXTURE reflecting an ancient bottleneck or relatively small population size in the past ($n = 5,833$ individuals). **b**, Sum of ROH per individual as a function of birth year ($n = 5,833$ individuals). Solid lines show ROH overall, and dashed lines indicate ROH overlapping ancestries from the Americas (AMR). ROH are divided into small, medium and large ROH, as in **a**. Smoothed conditional mean lines are shown using the locally estimated scatterplot smoothing method. Error bands represent 95% confidence intervals. **c**, Mutation burden in different ancestries shows the effects of bottleneck events in causing loss of rare variants ($n = 5,818$ individuals). Rare variants are correlated with levels of ancestries

from the Americas, Western Europe or West Africa for rare variants (derived allele frequency ≤ 5%). Smoothed conditional mean lines are shown using a linear model. Error bands represent 95% confidence intervals. Spearman correlation values are shown ($R$ and two-sided $P$ values) for all ancestries. Analysis of whole-genome sequences from 1000 Genomes MXL samples shows that the rare mutation burden result is robust to ascertainment bias of Illumina's Multi-Ethnic Global Array (Supplementary Figs. 39 and 40). Variants were annotated using the Variant Effect Predictor tool, and nonsynonymous (deleterious) variants are a combined set of missense variants predicted to be damaging by polyphen2 along with splice, stop lost and stop gained variants.

observed for intergenic variants) (Fig. 3c) in contrast to other ancestries. We verified these observations with whole-genome sequences from a 1000 Genomes Project cohort (Mexican Ancestry in Los Angeles, California or MXL) (Supplementary Fig. 39), as well as with 50 genomes sequenced as part of the MXB project (Supplementary Fig. 40), to rule out ascertainment biases due to the array genotyping. Our result probably reflects primarily founder events during the peopling of America or subsequent genetic drift leading to loss of rare variants and/or their rise to higher frequencies.

## GWAS and polygenic prediction in the MXB

To understand trait-associated locus transferability, we conduct GWAS analyses across 22 binary and quantitative traits (Supplementary Table 6). We identify genome-wide significant loci passing Bonferroni correction ($P < 2.27 \times 10^{-9}$) on chromosomes 1, 9, 11 and 16 associated with lipid levels in blood (Fig. 4a). Fine-mapping of independent signals within these loci reveals variants in or near *CELSR2*

(low-density lipoprotein (LDL): rs7528419), *ABCA1* (high-density lipoprotein (HDL): rs9282541 and rs2065412), the *LINC02702–BUD13–ZPR1–APOA1–APOA4–APOA5–APOC3–SIK3* locus (HDL: rs180326 and rs200905431; LDL: rs66505542; triglycerides: rs947989, rs66505542 and rs5104), *HERPUD1–CETP* (HDL: rs57502215, rs56129100, rs193695, rs56228609 and rs117427818; cholesterol: rs57502215, rs56228609 and rs118146573) and *APOE* (LDL: rs7412; triglycerides: rs440446), which have all previously been associated with lipid levels in European and Hispanic groups (Supplementary Table 7). Notably, we replicate the association of the *ABCA1\*C230* allele that has previously been associated with decreased HDL cholesterol levels ($\beta = -0.219$, s.e. = 0.030, $P = 1.64 \times 10^{-13}$; Fig. 4a), and is found almost exclusively in Indigenous groups from the Americas[43]. This association was replicated in the subset containing >90% inferred Indigenous ancestries although it did not reach genome-wide significance ($\beta = -0.210$, s.e. = 0.055, $P = 1.22 \times 10^{-4}$). Restricting the GWAS cohort to individuals with >90% inferred ancestries from the Americas did not identify any genome-wide significant loci.

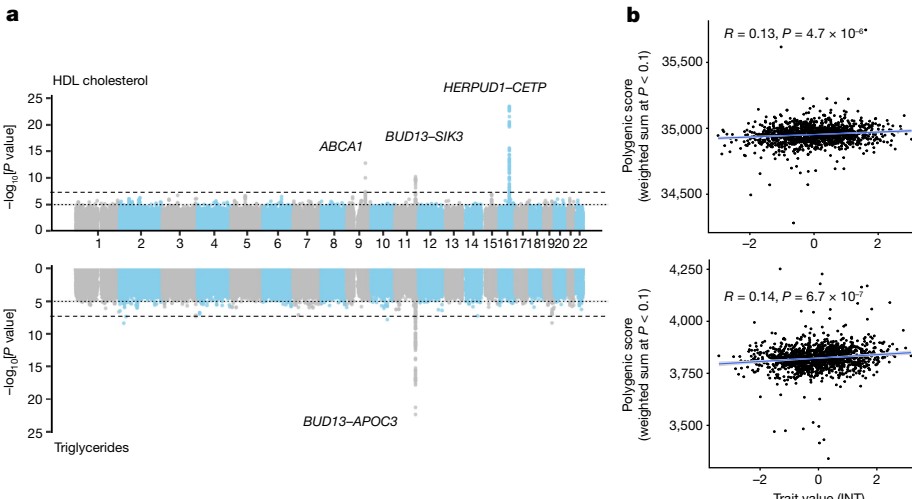

**Fig. 4 | Illustrative examples of GWAS and polygenic prediction in the MXB.**
**a**, Manhattan plots showing GWAS results for HDL cholesterol (top, $n = 4,484$)
and triglycerides (bottom, $n = 4,483$) in the full MXB dataset. Fine-mapped
genes are labelled (Methods). To aid with visualization, 1 in 200 SNPs with
$P > 0.01$ were sampled for the Manhattan plots. **b**, Prediction performance is
measured by the correlation between polygenic score (the sum of all alleles
associated at $P < 0.1$ weighted by their estimated effect sizes) and trait value
(as measured by Pearson correlation $R$ and its associated two-sided $P$ value)
for HDL cholesterol (top, $n = 1,327$) and triglycerides (bottom, $n = 1,326$).

According to the schematic in Supplementary Fig. 41, for **b**, GWAS was
carried out in two-thirds of the MXB, and the remaining one-third of the
MXB was used to compute polygenic scores and test their ability to predict
complex traits. Smoothed conditional mean lines are shown using a linear
model. Error bands represent 95% confidence intervals. Scores were
computed using TOPMed-imputed MXB genotypes. Traits were normalized
using an inverse normal transform (INT) for both **a** and **b**. For further evaluation of
prediction performance, see Extended Data Figs. 1b and 2–10 and Supplementary
Tables 8 and 9.

To assess transferability in the prediction of quantitative traits
using polygenic scores, we re-perform a GWAS in only 4,000 randomly
selected individuals from the MXB and construct polygenic scores in
the remaining 1,778 individuals (Supplementary Fig. 41). We compute
polygenic scores using both genotype data and imputed genotypes
using TOPMed. To assess the impact of using different GWAS sum-
mary statistics on prediction performance, for comparison we also
compute polygenic scores using pan-ancestry GWAS from the UKB in
light of the varying ancestry sources in Mexico (Fig. 4b and Extended
Data Fig. 1b). We observe that MXB-based prediction works better or
as well as UKB-based prediction, despite much lower sample size, for
glucose, creatinine, cholesterol and diastolic blood pressure (Extended
Data Figs. 1b and 2–10 and Supplementary Tables 8 and 9). Triglycer-
ides, HDL and LDL cholesterol levels are also almost as well predicted
by the MXB GWAS (Fig. 4b). These results indicate that further gains
in prediction power would be achieved by increasing the sample size
further. Although many factors are probably involved in differential
polygenic score portability by trait, some trait architecture features
are probably relevant, such as the strength of stabilizing selection
that the trait is under, its mutational target size and heritability per
causal site[44,45]. Using estimated mutational target sizes from previous
GWAS studies[45], we observe that traits with smaller inferred mutational
target sizes (creatinine and triglycerides) are predicted better with SNPs
discovered in MXB compared to traits inferred to have larger target
size (height and body mass index (BMI))[45]. UKB-based predictors are
used in our complex trait modelling below, as these can be computed
for all MXB individuals.

## Complex trait architectures in the MXB

Last, we assess the contribution of genetic variation resulting from
variable demographic and environmental histories or causal variant
distributions towards affecting variation in complex traits or diseases
in Mexico (Supplementary Fig. 42). We focus on several quantitative
traits: height, BMI, triglycerides, cholesterol, glucose, blood pres-
sure and others. Aiming to understand how the traits are distributed

geographically and relative to single model covariates, we first visualize
average trait values by units of our biogeographical and sociocultural
factors to understand the dimensions of trait variation (Fig. 5a and
Supplementary Figs. 43–51).

Next we use a mixed model to estimate the contribution of genetic
factors to trait variation jointly modelled with the environmental
factors (Fig. 5b,d and Supplementary Figs. 52–61). Genetic ancestry
proxies can be associated with complex traits due to genetic factors
or due to non-genetic factors that covary with genetic ancestries such
as differential experiences of discrimination, dietary nutrition and
socioeconomic status (Supplementary Fig. 42). The genetic factors
that vary with genetic ancestry proxies can be different distributions
of ROH or other differential patterns of genetic variation caused by
demographic and environmental histories that vary among ances-
tries. ROH have also previously been shown to have associations with
a broad range of complex traits such as height, weight and cholesterol,
pointing towards a recessive architecture of these traits[36,46]. As shown
above, genetic ancestry proxies in the MXB are correlated with the
number and length of ROH (Fig. 3a). We, therefore, develop a mixed
model for the association of genetic factors such as ancestry prox-
ies, ROH and polygenic scores with trait variation. We consider in our
model several environmental factors to improve power and to query
the role of genetic factors reflected in ancestry proxies compared
to environmental factors. We include variables available in the MXB
related to discrimination, socioeconomic opportunities and living
environment (collectively called sociocultural and biogeographical
factors), as well as unobserved random effects to model cryptic related-
ness and potential unmodelled environmental factors. In this model, a
significant association with ancestry proxies could reflect the associa-
tion of particular causal genotypes with those ancestries or associated
unmodelled environmental factors such as nutrition. Our combined
model explains 66.6% of the variance for height, 30.4% for BMI, 44.3%
for triglycerides, 30.9% for cholesterol and 30.91% for glucose.

As an illustrative example, height values show a clear increasing
pattern from southeast to northwest in the MXB (Fig. 5a). Even though
height values in every state exhibit a large variance (Fig. 5a), height is

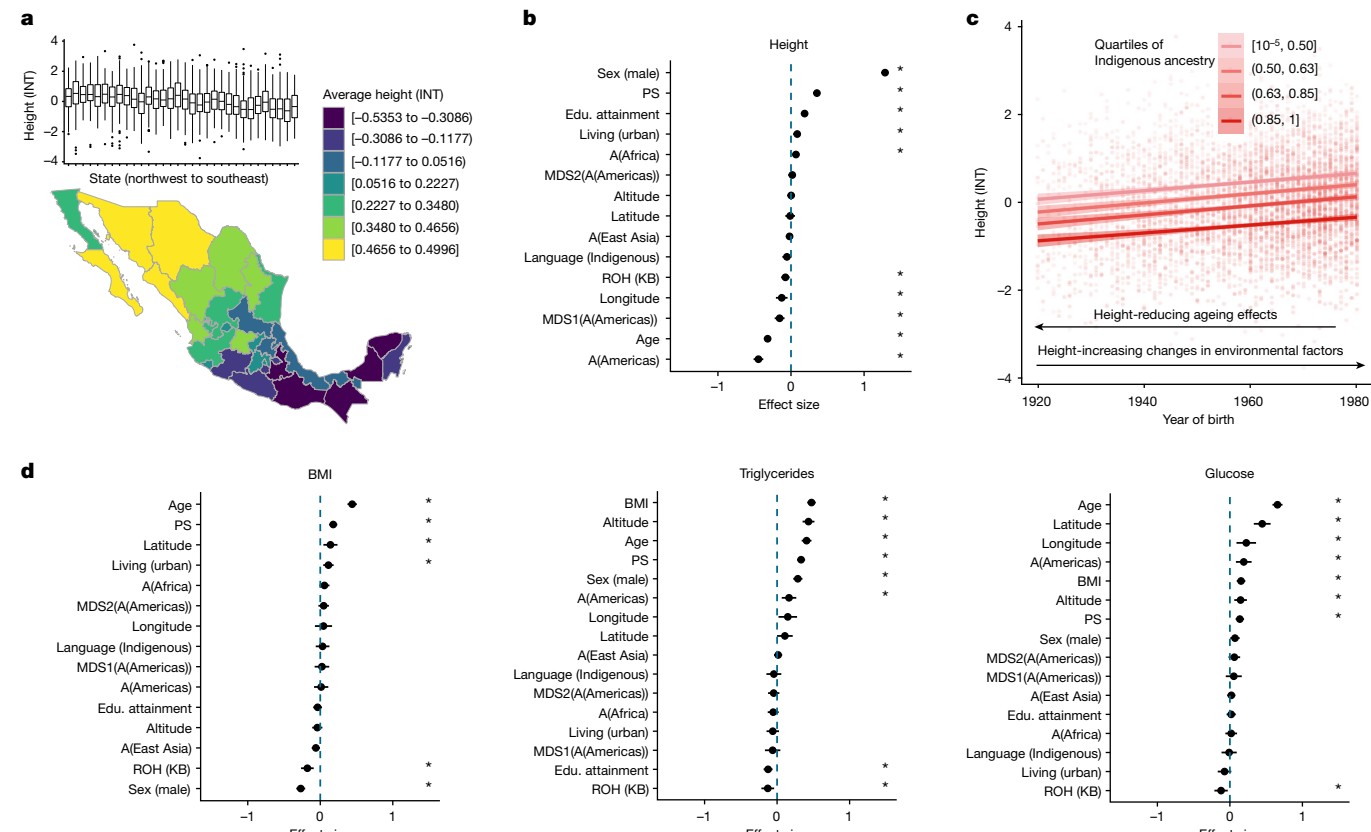

**Fig. 5 | An analysis of the factors influencing height and other complex trait variation. a**, Bottom: map of average height in Mexico (n = 5,770). Height was normalized using an INT. Top: box plots of height (INT) variation in each state from northwest to southeast. The box plots show the median value and the quartiles. Whiskers extend to the minimum and the maximum values. The dots represent outliers. n = 5,846 biologically independent samples were used for the analysis. **b**, Explanatory model for height variation implicates the role of genetics and environment. The plot shows effect-size estimates and confidence intervals (1.96 × s.e.m.) from a mixed-model analysis. All quantitative predictors are centred and scaled by 2 standard deviations. Asterisks show significance at false discovery rate < 0.05 across traits and predictors analysed[50]. n = 4,625 biologically independent samples were used for the analysis. **c**, Height as a function of birth year in quartiles of ancestries from the Americas (n = 5,598).

Error bands represent 95% confidence intervals. **d**, Trait profiles for BMI (left), triglycerides (middle) and glucose (right). Results of mixed-model analysis, as in **b**. The plot shows effect-size estimates and confidence intervals (1.96 × s.e.m.) from a mixed-model analysis. n = 4,607, 3,664 and 3,613 biologically independent samples were used for the analysis for BMI, triglycerides and glucose, respectively. For **b** and **d**, PS are polygenic scores computed using UKB summary statistics (SNPs significant at $P < 10^{-8}$), A(Africa/East Asia/Americas) refers to ancestry proportions from that region as inferred from ADMIXTURE, and MDS1(A(Americas)) and MDS2(A(Americas)) refers to multidimensional scaling (MDS) axes within ancestries from the Americas as inferred using a MAAS-MDS analysis (Supplementary Fig. 24). Educational (Edu.) attainment is on a scale from 0 to 8 (low to high educational attainment), and altitude is measured in metres (low to high).

significantly correlated with longitude (Fig. 5b and Supplementary Figs. 43 and 45). We find that individuals with a higher proportion of Indigenous ancestries from the Americas are significantly shorter ($\beta = -0.45, P < 2.2 \times 10^{-16}$) whereas individuals with a higher proportion of ancestries from West Africa are significantly taller ($\beta = 0.07, P < 0.005$; Fig. 5b). Further, considering ancestries at a finer resolution, we observe decreased height with a change in ancestries from the north of Mexico (for example, Huichol and Tarahumara) to those from the Mayan region (for example, Tojolabal and Maya; $\beta = -0.156, p = 6.13 \times 10^{-6}$; Supplementary Fig. 55). Total length of ROH is also significantly associated with shorter height ($\beta = -0.08, P = 0.01$). Simultaneously, younger individuals across the ancestry spectrum are taller than older individuals with the same ancestries (Fig. 5c), exhibiting the impact of non-genetic factors (improving nutrition in the birth year range studied or effects of ageing) on height variation as well.

Obesity is a public health issue in Mexico[47] and has been suggested to be related to higher genetic risk associated with Indigenous ancestries[48]. Contrary to this hypothesis, in the MXB as a whole, when considered univariately, Indigenous genetic ancestries and speaking an Indigenous language actually correlate with lower BMI (Supplementary Figs. 49 and 60). In our joint model with covariates, although those

associations disappear, ROH in a genome (which are more prevalent in Indigenous genetic ancestries) are also associated with lower BMI. By contrast, as living in an urban environment is associated with higher BMI (Fig. 5d), our results suggest a focus on factors related to an urban environment such as diet and sedentarism to help tackle the obesity issue in Mexico. Further segmented analysis considering only individuals in urban environments suggests the same: we observe individuals that speak an Indigenous language associating with higher BMI only in urban environments (Supplementary Fig. 61).

By contrast, some other traits show a correlation with an individual's proportion of inferred genetic ancestries from the Americas: creatinine ($\beta = -0.13, P = 0.0095$), LDL ($\beta = -0.141, P = 0.013$), triglycerides ($\beta = 0.16, P = 0.001$) and blood glucose level ($\beta = 0.19, P = 0.0005$; Supplementary Fig. 54). In the MXB, the amount of an individual's genome in ROH is associated with lower BMI ($\beta = -0.18, P = 7.11 \times 10^{-5}$), triglycerides ($\beta = -0.13, P = 0.004$) and blood glucose level ($\beta = -0.12, P = 0.01$; Supplementary Fig. 56). We also find that polygenic scores computed using genome-wide significant SNPs from the UKB pan-ancestry GWAS are a significant predictor for complex trait variation for all traits analysed (Supplementary Fig. 57). Blood pressure is associated with environmental factors and polygenic scores but not other genome-wide

genetic factors (Supplementary Fig. 53). Notably, among the environmental factors investigated, living in an urban environment is associated with higher height, BMI, cholesterol and creatinine levels, whereas living at high altitudes is significantly associated with higher triglyceride, glucose, cholesterol, creatinine and blood pressure levels (Supplementary Fig. 58). Higher educational attainment is associated with higher height, LDL, HDL and lower triglyceride levels, whereas speaking an Indigenous language is associated with lower creatinine and cholesterol levels (Supplementary Fig. 58).

Previous work has implicated the *ABCA1*C230* allele (rs9282541) in decreasing HDL levels and shown that this allele is apparently exclusive to Indigenous genetic ancestries from the Americas (found in 29 of 36 Native American groups, but not in European, Asian or African individuals)[43]. In the MXB, we similarly observe the *ABCA1*C230* allele to be in higher frequencies in individuals with a higher proportion of ancestries from the Americas, and observe that individuals with higher *ABCA1*C230* allele frequencies have lower HDL levels (Supplementary Fig. 59a). Nevertheless, overall, Indigenous ancestries are not associated with HDL levels after accounting for other covariates. In fact, genetic variants collectively on the Indigenous genetic background are associated with lower LDL levels (Supplementary Fig. 59b). These results illustrate how the interplay between cultural and diet factors and genetic factors are essential for different cholesterol outcomes. They also imply that although some functional variants may be specific to regions or genetic backgrounds, these are few (about 1,000 such variants estimated in the Americas with a frequency of 40% or higher from the Human Genome Diversity Project sampling of diverse genomes[49]), and caution against using an individual's global ancestry proportion as a predictor of the effect of a single functional variant. Our results overall support that functional variants with variable frequencies or environmental interactions are partially responsible for variation in a range of complex traits in Mexico[43].

## Conclusion

Our work demonstrates the value of generating genotype–phenotype data on underrepresented groups to reveal lesser-known genetic histories and generate findings of biomedical relevance. It is also an illustration of the joint modelling of genetic and environmental effects to reveal the aetiology of complex traits and disease. In this project, we ensure diverse Indigenous and rural presence in our sampling strategy, consider the fluidity of ancestries from different local and global regions in our analyses, and evaluate their reflection in genetic and disease-relevant complex trait variation. By leveraging the largest nationwide genomic biobank in Mexico, we find diverse sources of ancestries in Mexico in light of its unique history, and infer demographic and admixture histories and ROH using ancestry-specific haplotype identity that reveal an elaborate fine-scale structure in the country. Observing a larger number of small ROH in younger individuals in the MXB and in genomic segments of Indigenous ancestries is relevant for parsing the genetic architecture of complex traits and diseases, especially those with a recessive component. We also show that demographic history affects the frequency distribution of genetic variants, thus changing how many rare variants individuals with different ancestries carry. We demonstrate the value of GWAS carried out on a resource such as the MXB for predicting complex traits. The MXB GWAS exhibits utility for polygenic score computation in independent Mexican cohorts, as well as for meta-analysis with other GWAS cohorts to increase prediction power further. Last, we observe a significant impact of genetic ancestries at different timescales, ROH, polygenic scores and sociocultural and biogeographic variables on various complex traits implicating the importance of both genetic and environmental factors in explaining complex trait variation and in considerations of potential public health interventions. Our results exhibit the added importance of considering genetic factors for preventive and personalized medicine above and beyond environmental factors. Our results will inform the design of future genetic and complex trait studies in Mexico and Latin America, and will hopefully motivate additional efforts to strengthen local research capacity across Latin America and benefit underserved groups globally.

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

## Methods

### Encuesta Nacional de Salud 2000

Since 1988, Mexico has established periodical National Health Surveys (Encuesta Nacional de Salud (ENSA), originally conceived as National Nutrition Surveys) for surveillance of Mexican population-based nutrition and health metrics. In this study, we use data and samples collected from the survey carried out in 2000, the ENSA 2000. This survey was a probabilistic, multi-stage, stratified, cluster household survey conducted by the Mexican Secretariat of Health from November 1999 to June 2000. Research design and methods have been described elsewhere[51]. Participants were randomly selected to be representative of the civilian, non-institutionalized Mexican population at the state and national levels. Trained personnel conducted the interviews. Information was collected on household and sociodemographic characteristics, current health status, healthcare service usage and behavioural aspects of participants. Sera and buffy coats were obtained from 43,085 individuals aged 20 years or older. More than fifty publications have arisen from this survey providing critical insights into the status of national health alongside some genetic traits of the sampled population[52]. In particular, the inclusion of individuals from remote and rural locations in Mexico makes this survey unique. Given its large volume, sophisticated sampling design, breadth of demographic sampling and extensive trait data, the ENSA 2000 represents a valuable untapped genetic resource to link genetic markers and health outcomes.

### Phenotype, lifestyle and environmental data for the MXB Project

For each individual, we have access to a range of anthropometric, disease, lifestyle and environmental data. These variables are summarized in Supplementary Table 2. Serum samples were further used to measure a number of biochemical traits analysed in this study. All traits analysed in the complex trait analysis were preprocessed as follows.

Biometric data were filtered to remove outliers with apparent errors in data entry. Outliers were identified on the basis of distribution density over the complete dataset of >6,000 individuals, resulting in height between 100 and 200 cm and weight between 25 and 300 kg.

Biochemical traits were similarly curated to remove extremes and negative values (<0). Glucose was also checked against finger prick tests taken at the time of the survey, and values that were greatly discordant were also removed. Glucose measurements were further stratified by random or fasting glucose samples based on participant questionnaire responses.

Blood pressure was manually curated for individuals for whom values differed by more than 20 units for the two readings taken, for whom diastolic pressure was higher than systolic, or for whom values were unusually high or low (<30 or >300). In these cases, both readings were manually checked, and discordant readings were discarded. These updated values were then merged with the remaining samples. A set of adjusted blood pressure phenotypes was also generated, adjusting for treatment for hypertension. In those individuals who were reported to be receiving some form of hypertension treatment, 15 units were added to systolic blood pressure and 10 to diastolic blood pressure (SBP_adj and DBP_adj)[53,54].

Quantitative traits were normalized using an inverse normal transform before complex trait analyses.

For each individual, we have access to data for various sociocultural factors such as access to healthcare and clean water, yearly income, educational attainment, whether they speak an Indigenous language or not, and whether they live in a rural or urban environment.

Localities were assigned values of latitude, longitude and altitude (metres) using data from the National Institute of Statistics and Geography (INEGI) in Mexico.

### Sample selection and genotyping for the MXB Project

To select the subset of biobanked samples to be genotyped, we first identified the total number of localities represented in the collection of extracted DNAs (that is, 898 recruitment sites). We then allocated one sample to each locality in consecutive additive rounds targeting an average sample size of 5 to 10 individuals regardless of population density. The initial rounds were enriched for individuals who reported to speak an Indigenous language, and then randomly selected samples were included until saturating budget capacity. This strategy ensured maximization of both geographic coverage and representation of Indigenous ancestries, resulting in a total of 6,144 samples distributed nationwide. A further subset of 87 samples failed DNA quality control or hybridization during genotyping, for a total of 6,057 successfully genotyped samples. Samples were genotyped on the Illumina's Multi-Ethnic Global Array (MEGA). The design of this array was previously led by C.R.G. and G.L.W. Several properties place the MEGA array as the ideal choice for biobank genotyping. It captures 1,748,250 SNPs derived from admixed population studies, making it broadly applicable in diverse populations. The array has boosted SNP coverage in both the MHC and KIR loci, a marker set of more than 30,000 SNPs for ancestry estimation, and includes more than 17,000 medically relevant genetic variants from previous GWAS and clinical studies. Such breadth of coverage of genomic diversity provides a comprehensive quantitative resource of the genetic variability in this cohort.título.

### Generation and quality control of MXB genetic data

Genome Studio was used to convert raw image files to plink files with raw genotype information. All SNPs were flipped to the forward strand, and duplicate SNPs were removed. For sites with missing chromosome number, physical position or both, we updated the map using the information in the SNP name or by mapping their rsID using dbSNP Build 151.

We removed all individuals with >5% missing genotype data and all genotypes with >5% missing individuals. We restricted the analyses to autosomes and removed all monomorphic SNPs. We restricted the analysis to biallelic SNPs and removed all SNPs with an ambiguous strand for all downstream analyses. All related individuals were detected using plink (--Z-genome --min 0.5) after pruning for linkage disequilibrium (--indep-pairwise 50 5 0.5). A script was written to iteratively find and remove related individuals to obtain the final quality-controlled dataset.

### Sources and quality control for reference panels

Reference genetic panels were used for various analyses of population structure. We used global populations from the 1000 Genomes Project (1KGP)[40] and the Human Genome Diversity Project (HGDP)[55], Zapotec individuals from Oaxaca from the Population Architecture using Genomics and Epidemiology Study (PAGE)[56], and Indigenous individuals from across Mexico from the Native Mexican Diversity Project (NMDP)[12] for the analyses of population structure and ancestry.

For each reference panel, we restricted the analysis to autosomes, removed all monomorphic SNPs, flipped all SNPs to the forward strand, and removed SNPs with an ambiguous strand.

### Anthropological classification

We used an anthropological and archaeological context to delineate different Mesoamerican regions[10]. An individual's locality was used to place them into one of the seven regions: the north of Mexico, the north of Mesoamerica, the centre, occident and Gulf of Mexico, Oaxaca and the Mayan region in the southeast[10]. This classification was used to visualize and regionalize some of the population structure and history analyses, especially those relating to Indigenous genetic substructure within Mexico.

### Note on genetic ancestries

Genetic ancestry arises from a set of paths through the ancestral recombination graph[57]. In this study, we obtain proxies for genetic ancestries using ADMIXTURE[20] (see below). As such, we are discretizing a continuous quantity for the purposes of understanding the effects of varying demographic histories on genetic and complex trait variation in MXB. The labelling and use of such discretized ancestry proxies remains a contentious issue[58]. To clarify the point that such proxies are not essentialized entities in the real world, but rather variables we use for the purposes just described, we opt to refer to our ancestry proxies as being from the region whose present-day individuals such proxies cluster with. Thus, we use, "ancestries from the Americas", "ancestries from West Europe", "ancestries from West Africa", "ancestries from South Asia" and "ancestries from East Asia" in the text, and shorter versions of the same for some figures (A(Americas) and so on).

Such regions are useful for our analyses only in so much as they reflect demographic and environmental histories that may affect the genetic and complex trait variation we are interested in. This is only one arbitrary scale to discretize at, and we also consider the origins and implications of ancestral variations within such regional groupings in several analyses, in which we carry out dimensionality reduction within such regional groupings (for example, MDS1(A(Americas)) and MDS2(A(Americas))).

Although not intended, the groupings used may seem to some as similar to racial categories that were created in the past 500 years and used to justify European superiority and colonization of global regions including present-day Mexico[58–60]. In Mexico, such categories have a similar history of racism and eugenics as in other parts of the world[61]. We reject fixed hierarchical categorizations of humans, as well as their use to justify the superiority of one group over another. We use ancestry proxies that are estimated from ADMIXTURE using unsupervised clustering, as well as axes of ancestry that result from dimensionality reduction within these ancestries, capturing variation among groups from the Americas, for example. Despite the confluence of genetic ancestries from around the globe in present-day Mexico, genetic ancestries in humans are continuous over time and space and should be considered only in that complexity and at different scales.

### Population structure analyses

For the analyses of population structure, we merged the quality-control-filtered MXB dataset and reference panels using plink. We repeated some of the quality control steps on the merged dataset, removing any monomorphic or duplicate SNPs. We also removed individuals with >5% missing genotype data, and genotypes with >5% missing individuals to obtain the clean merged dataset.

We carried out two sets of principal components analysis (PCA) and ADMIXTURE[20] analysis. One was carried out on the merged dataset including MXB, Zapotecs from the Population Architecture using Genomics and Epidemiology Study, and global populations from the 1000 Genomes Project and the Human Genome Diversity Project (Fig. 1b, Supplementary Figs. 6, 11 and 12 and Supplementary Table 3), and the other was carried out on the merged dataset including only MXB and individuals indigenous to present-day Mexico from the NMDP (Supplementary Figs. 8–10 and 13). $F_{ST}$ analysis was carried out on all MXB individuals, as well as on only MXB individuals with 90% ancestry from the Americas as estimated from the ADMIXTURE analysis (Supplementary Figs. 15–18).

smartpca from Eigenstrat[62] was used to carry out the PCA. Principal components generated by smartpca (Supplementary Figs. 6 and 7) were used to carry out the uniform manifold approximation and projection (UMAP) analysis (Fig. 1c and Supplementary Figs. 19 and 20)[63]. $F_{ST}$ analysis was carried out using smartpca.

Given the large loss of SNPs due to admixture linkage disequilibrium in our admixed Mexican individuals, we opted not to prune for

linkage disequilibrium for the population structure analyses presented in this study. We repeated the analysis on a set of SNPs pruned for linkage disequilibrium and obtained similar results (data not shown). Unless otherwise noted, given the admixed nature of the Mexican individuals, we did not remove SNPs owing to departure from Hardy–Weinberg equilibrium in the MXB, as many SNPs are expected to be out of Hardy–Weinberg equilibrium owing to admixture and population structure.

We also computed and visualized population structure using the method of ref. 5 ('archetypal analysis') with individuals from the quality-controlled MXB dataset and individuals from the 1000 Genomes, the Human Genome Diversity Project and the Population Architecture using Genomics and Epidemiology Study as our reference panel (Supplementary Figs. 21–23). We also carried out the analysis using only the quality-control-filtered MXB dataset. In both analyses, PCA results were generated only once and used as input to compute archetypes from $K = 3$ to 10. In reporting the results, we refer to the 'archetypes' in the analysis as 'sources', given that the word archetypes has connotations of pure types that are not necessary for the model to be applied to population genetic data.

### Analyses of subcontinental ancestry

Analyses were carried out to obtain axes of genetic variation or ancestry among a continental group. Such analyses also help interpret the specific origins of an ancestry present in Mexico today. These analyses were carried out using rfmix[2] to estimate local ancestry along the genome and pcamask[19] to carry out an ancestry-specific PCA for ancestries originating from present-day Africa. During the course of this study, new and improved methods to estimate local ancestry along the genome (GNOMIX)[3] and to carry out ancestry-specific PCA (Multiple Array Ancestry Specific Multidimensional Scaling, MAAS-MDS, an MDS designed for analysing samples from several different genotyping arrays simultaneously)[64] were published, allowing us to use these tools for the analysis of ancestry variation within the Americas for the complex trait analysis.

**MAAS-MDS on ancestries from the Americas.** For the MAAS-MDS[64] analyses, we used GNOMIX[3] for local ancestry inference using its preset 'best' mode and then masked the non-Indigenous segments. For the European reference, we used the cohorts Iberian populations in Spain (IBS) and British from England and Scotland (GBR) from 1KGP (198 samples)[40], for ancestries from Africa, the Yoruba in Ibadan, Nigeria (YRI) cohort from 1KGP (108 samples), and for ancestries from the Americas, Peruvian in Lima, Peru (PEL) from 1KGP (only those samples with >95% ancestry from the Americas) and the 50 genomes of Indigenous individuals across Mexico generated as part of the MXB Project (79 samples)[65]. For the PEL, we used an unsupervised clustering analysis with ADMIXTURE ($K = 3$) together with IBS and YRI from the 1KGP to find those PEL samples with >95% assignment to a cluster not shared with IBS or YRI; that is, with >95% ancestries from Americas. The additional 50 genomes from MXB were selected to have high Indigenous ancestries as described previously[65]. The reference genomes were merged with each array resulting in 856,352 SNPs in array 1 and 967,338 in array 2. Array 1 included 10 Indigenous groups from NMDP genotyped with the Affymetrix 6.0 array: Tarahumara, Huichol, Purepecha, Nahua, Totonac, Mazatec, Northern Zapotec (from Villa Alta district, Northern Sierra in Oaxaca state), Triqui, Tzotzil and Maya (from Quintana Roo state). Array 2 included the 6,051 individuals from the MXB project genotyped with MEGA. The MAAS-MDS was applied to the Indigenous American ancestry segments (that is, masking intercontinental components of African and European origin) in both arrays 1 and 2. The analysis was run using average pairwise genetic distances and considering only individuals with >20% Indigenous American ancestry, to generate ancestry-specific MDS axes for ancestries from the Americas in the MXB (Supplementary Fig. 24).

**asPCA on ancestries from Africa.** We carried out this analysis on all individuals in the MXB with ≥5% ancestry from Africa estimated from the admixture analysis. This resulted in 1,965 individuals with ancestry originating from present-day Africa. In this set of individuals, we used populations from the 1000 Genomes Project (CEU: Utah residents (CEPH) with Northern and Western European ancestry and YRI: Yoruba in Ibadan, Nigeria)[40] and the Population Architecture using Genomics and Epidemiology Study (Zapotecs from Oaxaca)[56] to estimate local ancestry using rfmix[2]. The MHC region was excluded from the analysis. SNPs out of Hardy–Weinberg equilibrium were removed from each of the reference panels ($10^{-3}$) and the MXB AFR ($10^{-8}$) subset beforehand. This dataset was merged with a subcontinental reference panel covering a range of groups in present-day Africa[21]. Pcamask[19] was used to mask all ancestries other than ancestries from Africa, and to generate ancestry-specific principal components for ancestries from Africa in the MXB (Supplementary Fig. 14).

## Population history analyses

**Ancestry-specific estimation of effective population size trajectories.** Analyses of population history using an approach that uses ancestry-specific identity-by-descent (IBD) segments were carried out on the entire MXB dataset, and on individuals belonging to each of the Mesoamerican regions (Fig. 2 and Supplementary Figs. 25–29). IBD segments of the genome can be used to estimate effective population size ($N_e$) for thousands of years into the past[30]. These IBD segments can be further overlapped with local ancestry tracts to obtain ancestry-specific IBD tracts to estimate population size in an ancestry-specific manner for an admixed cohort (this approach has been called asIBDNe)[4].

For this analysis, the MXB was merged with populations from the 1000 Genomes Project (CEU: Utah residents (CEPH) with Northern and Western European ancestry and YRI: Yoruba in Ibadan, Nigeria)[40] and the Population Architecture using Genomics and Epidemiology Study (Zapotecs from Oaxaca)[56]. SNPs in each population were previously filtered for Hardy–Weinberg equilibrium ($10^{-5}$ for reference groups and $10^{-10}$ for the MXB samples). The MHC region was excluded from the analysis. We repeated some of the quality control steps on the merged dataset, removing any monomorphic or duplicate SNPs. We also removed individuals with >5% missing genotype data, and genotypes with >5% missing individuals to obtain the clean merged dataset.

We followed a computational pipeline recommended by the developers of asIBDNe to call IBD segments and local ancestry along the genome. We used beagle (beagle.25Nov19.28d.jar)[66] to phase the data, refined-ibd (refined-ibd.17Jan20.102.jar)[67] to call IBD and merge-ibd-segments (merge-ibd-segments.17Jan20.102.jar) to remove breaks and short gaps in IBD segments, removing gaps between IBD segments that have at most one discordant homozygote and that are less than 0.6 cM in length. Local ancestry was estimated using rfmix. The rfmix output was rephased to match the original phasing. asIBDNe (ibdne.19Sep19.268.jar) was run to estimate ancestry-specific population sizes using a 2-cM IBD length threshold.

**AdmixtureBayes.** In this study, we used AdmixtureBayes[6] to generate, analyse and plot admixture graphs for a sample of 6,011 individuals from the MXB (Extended Data Fig. 1a and Supplementary Fig. 30). Our focus was on inferring the demographic history of Indigenous groups in Mexico, so we used only the allele frequencies of the Indigenous portions of the MXB genomes. In particular, we used GNOMIX for local ancestry inference as described in the section entitled 'MAAS-MDS on ancestries from the Americas' in the Methods, and masked the non-Indigenous segments.

We grouped the individuals on the basis of Mesoamerican regions of Mexico, to understand the variation of Indigenous demographic histories across the country. We used Han Chinese as an outgroup for the Indigenous ancestries.

Using AdmixtureBayes, we inferred the split events and admixture events that have occurred in the MXB. We used the default parameters for generating the admixture graph with the exception of the number of chains and iterations, which we set to a higher value of 16 (--MCMC_chains 16) and 20,000 (--n 20000) to ensure convergence; we also used the -slower flag, enabling the computation of the necessary information to plot the top trees, and a burn-in period corresponding to half the samples. We plotted the tree with the highest posterior probabilities, which provides a visual representation of the inferred admixture events and allows us to explore the uncertainty in the inferences. Further details of the AdmixtureBayes method and prior used can be found in the corresponding paper[6].

## ROH

The MXB dataset was pruned for linkage disequilibrium using plink (--indep-pairwise 50 5 0.9). ROH were estimated using plink (--homozyg) identifying 349,400 ROH. We estimated the number of ROH carried by an individual (nROH) and the total sum of ROH in an individual in kilobases (sROH or sumROH) (Fig. 3 and Supplementary Figs. 31–33). ROH were divided into small, medium and large according to the theoretical framework in ref. 37. Python scripts were used to categorize ROH by length, and to overlap ROH with local ancestry calls from rfmix to obtain ancestry-specific ROH summary statistics (Supplementary Table 5). Local ancestry calls were the same as those used for the asIBDNe analysis. A total of 38,340 ROH did not overlap a homozygous local ancestry assignment and were removed from this analysis; the remaining 311,060 that overlapped a homozygous local ancestry assignment were kept. We used a python script to compute the number of ROH in ancestry switch points as well (58 ROH or 0.00019 of all ROH fell within an ancestry switch and were also excluded from the analysis).

ROH were also correlated with birth year in the MXB (Fig. 3b) and used as a variable in the complex trait mixed-model analysis. For the birth year analysis, we removed the first two decades, as each year has below 15 individuals sampled in this period. Birth year was also directly correlated with ancestries from the Americas (inferred using ADMIXTURE) in rural and urban localities separately. ROH were also correlated with global ancestries per individual estimated from the admixture analysis (Fig. 3a and Supplementary Fig. 32). An R script was used to analyse distributions of the sum of ROH by geography (Supplementary Figs. 31 and 33).

## Mutation burden analyses

Variants were annotated according to whether they were ancestral or derived, and their functional effect depending on their location in a gene or genome. Ancestral alleles for each SNP in the MXB were inferred using the EPO pipeline from the 1000 Genomes Project. Variant Effect Predictor[68] was used to annotate the effect of a variant using the humdiv database, and picking one consequence (or transcript) per variant according to a criterion that includes the canonical status of the transcript, APPRIS isoform annotation, transcript support level, biotype of transcript ('protein_coding' preferred) and consequence rank preferring high impact.

Mutation burden is defined as the sum of derived alleles carried by an individual. A computational pipeline using vcftools, python, linux and R was used to compute mutation burden in different classes of variants, and at different derived allele frequency thresholds. We computed either a rare mutation burden (derived allele frequency ≤ 5%) or an overall mutation burden considering all allele frequencies. Our pipeline used the R packages matrixStats, dplyr and ggplot2. We correlated the mutation burden with the global ancestry percentage from different present-day continental origins in all individuals. The ancestry estimates were from the admixture analysis. We computed a Spearman's correlation and $P$ value (Fig. 4).

This analysis was repeated in the 1000 Genomes Project Mexican Ancestry in Los Angeles, California (MXL) cohort (Supplementary

Fig. 39). This was to check whether the effect we were observing was due to ascertainment bias in the MEGAex array that covers fewer rare variants predominantly native to the area that is Mexico today. The whole-genome sequences from the 1000 Genomes Project allowed us to rule this out. Ancestry estimates were generated using ADMIXTURE with reference panels from 1000 Genomes (CEU: Utah residents (CEPH) with Northern and Western European ancestry, GBR: British in England and Scotland, YRI: Yoruba in Ibadan, Nigeria and PEL: Peruvian in Lima, Peru) and 50 whole-genome sequences of Indigenous individuals across Mexico generated as part of the MXB Project[65]. Variant effect predictor was used to annotate SNPs, and mutation burden was computed in the same manner. The deleterious category includes the following consequence terms: splice acceptor variant, splice donor variant, stop gained, stop lost and start lost.

## GWAS analyses

**Phenotype definitions and quality control.** Binary health-related phenotypes were defined on the basis of questionnaire responses. Cases were defined on the basis of a positive response to the questionnaire questions. Controls were those who responded with 'no'. Individuals responding with 'do not know', 'prefer not to answer' or 'no response' were excluded (Supplementary Table 6). Additionally, arthritis cases were defined as any individual with gout arthritis, rheumatoid arthritis and/or other forms of arthritis. Two hypertension phenotypes were defined: Hypertension_1, based on a diagnosis of hypertension; and Hypertension_2, which additionally took into account blood pressure readings. Cases were defined on the basis either a diagnosis for hypertension, medication or blood pressure readings greater than 140/90.

Quantitative traits were measured as previously described[51]. Data were filtered to remove outliers with apparent errors in data entry, and negative values (<0) based on distribution density over the dataset. Height was limited to participants with measurements between 100 and 200 cm; weight was restricted to between 25 and 300 kg. Glucose and fasting glucose levels were checked against finger prick tests taken at the time of the survey and values that were greatly discordant were removed. Fasting glucose measurements were defined on the basis of whether participants had eaten in the 8–12 h before the samples being taken.

Blood pressure was manually curated for individuals for whom values differed by more than 20 units for the two readings taken, for whom diastolic pressure was higher than systolic, or for whom values were unusually high or low (<30 or >300). In these cases, both readings were manually checked, and discordant readings were discarded. These updated values were then merged with the remaining samples. For GWAS, the first set of readings was used unless removed during the quality control process, in which case the second set of readings was used, if available. A set of adjusted blood pressure phenotypes was also generated, adjusting for treatment for hypertension. In those individuals who were reported to be receiving some form of hypertension treatment, 15 units were added to systolic blood pressure and 10 to diastolic blood pressure.

**GWAS.** GWAS analyses for both binary and quantitative traits were carried out with regenie (v3.1.3)[69]. Before GWAS, individuals with mismatched sex or IBD > 0.9 were removed. Quantitative traits were inverse normalized before analysis. Only case–control traits with more than 100 cases were taken forward for analysis. For all analyses, age, sex and the first four principal components were included as covariates. For cholesterol, triglycerides, HDL, LDL, hypertension and fasting glucose, BMI was also included as a covariate.

**Polygenic score GWAS.** GWAS was carried out on a random subset of 4,000 individuals with genotype data available, as described above. For quantitative traits, raw values were again normalized within the selected subset before analysis.

**Fine mapping of GWAS-significant loci.** Lead association SNPs and potential causal groups were defined using FINEMAP (v1.3.1; $R^2 = 0.7$; Bayes factor ≥ 2) of SNPs within each of these regions on the basis of summary statistics for each of the associated traits[70]. FUMA SNP2GENE was then used to identify the nearest genes to each locus on the basis of the linkage disequilibrium calculated using the 1000 Genomes EUR populations, and explore previously reported associations in the GWAS catalogue[40,71] (Supplementary Table 7).

## Polygenic score analyses

We computed polygenic scores using plink and summary statistics from the MXB GWAS conducted on 4,000 individuals as described above[72]. We computed scores on the remaining 1,778 individuals. We also computed scores for the same individuals using pan-ancestry UKB GWAS summary statistics (https://pan.ukbb.broadinstitute.org)[7,8] (Supplementary Fig. 41). Linkage disequilibrium was accounted for by clumping using plink using an $r^2$ value of 0.1, and polygenic scores were computed using SNPs significant at five different $P$-value thresholds (0.1, 0.01, 0.001, 0.00001 and $10^{-8}$) with the --score sum modifier (giving the sum of all alleles associated at a $P$-value threshold weighted by their estimated effect sizes). We tested the prediction performance of polygenic scores by computing the Pearson's correlation between the trait value and the polygenic score (Supplementary Tables 8 and 9). Further, we created a linear null model for each trait including age, sex and ten principal components as covariates. We created a second polygenic score model adding the polygenic score to the null model. We computed the $r^2$ of the polygenic score by taking the difference between the $r^2$ of the polygenic score model and the $r^2$ of the null model. In general, MXB-based prediction is improved by using all SNPs associated at $P < 0.1$ and using TOPMed-imputed data, whereas the UKB-based prediction shows its best performance using only genome-wide significant SNPs (at $10^{-8}$ or $10^{-5}$) and only genotyped data (Extended Data Fig. 1b and Supplementary Tables 8 and 9).

## Complex trait variation models

To assess the factors involved in creating complex trait variation, we carried out a mixed-model analysis using the lme4qtl R package for all quantitative traits. lme4qtl allows flexible model creation with multiple random effects[73].

We considered several genetic and environmental variables as fixed predictors of complex trait variation. Genetic variables included polygenic scores computed using UKB summary statistics (SNPs significant at $P < 10^{-8}$) for each trait, genetic ancestries estimated from ADMIXTURE, continuous axes of ancestry variation estimated using MAAS-MDS, and ROH (amount of ROH carried in an individual genome in kilobases). We also considered biogeographical variables such as latitude, longitude and altitude (metres). We considered demographic variables of age and sex. Last, we considered sociocultural variables: educational attainment (which shows a positive correlation with income levels (Supplementary Fig. 52); however, income levels are available only for a third of the individuals); whether they speak an Indigenous language or not as a proxy for differential experience of discrimination and culture; and whether they live in an urban or rural environment. BMI was included as a covariate for all quantitative traits except height, BMI and creatinine (Fig. 5 and Supplementary Figs. 53–58). To ease interpretation of the mixed-model coefficients for jointly considered numeric and binary predictors, we standardized predictor variables as follows[74]. To make coefficients of numeric predictors comparable to those for untransformed binary predictors, we divide each numeric variable by two times its standard deviation[74]. We centred both the binary and numeric predictors. All of the covariates mentioned above are significant when jointly modelled for at least one tested trait, justifying their use in the full model.

We also include two random predictors in our model. These are: the covariance structure defined by the genetic relationship matrix; and

the locality where the individual is from to capture any other environmental variation (such as diet) not captured by the fixed predictors.

The genetic relationship matrix was generated using the GENESIS R package using kinship coefficients. As kinship estimates can be inflated under the presence of population structure and admixture, we obtained kinship coefficients for the genetic relationship matrix in the following manner: (1) PC-air[75] was used to obtain principal components that capture ancestry and not relatedness (this procedure used kinship coefficients estimated using KING[76] as input to partition samples into a related (5,562) and unrelated (271) set (using kinship threshold 0.044) and carrying out PCA on the unrelated set); (2) PC-relate[77] was used to obtain kinship coefficients that capture relatedness but not ancestry (this method uses the ancestry-representative principal components from (1) to correct for population structure before calculating the kinship coefficients).

For this analysis, we removed rare variants (MAF < 5%), regions with known long-range linkage disequilibrium[78,79] and variants in high linkage disequilibrium ($r^2 > 0.1$ in a window of 50 kb and a sliding window of 1 variant).

To account for multiple significance testing, the false discovery rate was controlled at 0.05 using the approach of Benjamini–Hochberg[50].

ABCA1 variant frequencies were computed using plink in individuals from the MXB stratified by ancestry proxies from ADMIXTURE or by HDL cholesterol levels (Supplementary Fig. 59).

Maps of Mexico to visualize trait distributions were created using the mxmaps R package (Supplementary Fig. 43). Variog from the GeoR R package was used to compute variograms on complex traits, with longitude and latitude used to compute distance (Supplementary Fig. 51).

## Inclusion and ethics

Samples were collected as part of the 2000 National Health Survey (ENSA 2000) conducted by the National Institute of Public Health (Instituto Nacional de Salud Pública (INSP)) across Mexico. The ENSA 2000 was carried out following the strictest ethical principles and in accordance with the Helsinki Declaration of Human Studies. Informed consent was obtained from all participants after extensive community engagement. National Health Surveys have been conducted periodically in Mexico since 1988, so the community is engaged with the study and receptive to household visits by INSP staff and fieldwork teams. As described in the original methodology[51], the ENSA 2000 involved a 2-h visit to each household. Before recruitment, the team met with the political, religious and community leaders of each locality to communicate the nature of the study, answer all questions and engage with the community. This community engagement process was essential in every recruitment site, with an emphasis on Indigenous and rural communities to ensure understanding of the study. Extracted DNAs have been stored and maintained at the INSP (Cuernavaca, Mexico), and selected samples were genotyped at the Advanced Genomics Unit of CINVESTAV (Irapuato, Mexico) through a collaboration agreement. The data have been jointly analysed, promoting local leadership and participation of Mexican researchers and trainees. The project was reviewed and approved by the Research Ethics Committee and the Biosafety Committee of the INSP (Institutional Review Board approvals CI: 1479 and CB: 1470). For the present project, personally identifiable data were removed from the dataset.

## Reporting summary

Further information on research design is available in the Nature Portfolio Reporting Summary linked to this article.

## Data availability

The genotype and phenotype datasets for the 6,057 newly genotyped individuals from the MX Biobank Project are available at the European Genome-phenome Archive (EGA) through a Data Access Agreement with the Data Access Committee (EGA accession number for study: EGAS00001005797; datasets: EGAD00010002361 (Mexican_Biobank_Genotypes) and EGAD00001008354 (Mexican Biobank 50 Genomes)). Data can be accessed only for academic research and non-commercial use. GWAS summary statistics generated as part of this study are available at https://doi.org/10.5281/zenodo.7420254.

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

**Acknowledgements** We thank the participants of ENSA 2000 (2000 National Health Survey), conducted in Mexico nationwide by the Secretaría de Salud (Health Secretariat) and the INSP; M. Hernández and C. Alpuche Aranda for institutional support from INSP; M. Flores, R. Nájera and A. Garmendia for project management support; C. Conde, V. Guerrero Lemus, A. Mendez Herrera, C. Portugal García, R. Rodríguez and M. Velázquez Mesa for biobank maintenance and sample preparation; M. Ortega, C. Gutiérrez and S. García for technical assistance; J. Cervantes for information technology support; H. Ringbauer for advice on the ROH analysis; M. Levin for assistance in archetypal analysis visualization; J. E. Rodríguez for conversations about population structure in Mexico; and A. Zaidi for comments on an earlier draft of this manuscript. This work was mainly supported by The Mexican Biobank Project: Building Capacity for Big Data Science in Medical Genomics in Admixed Populations, a Mexico–UK binational initiative co-funded equally by CONACYT (grant number FONCICYT/50/2016), and The Newton Fund through The Medical Research Council (grant number MR/N028937/1) awarded to A.M.-E. and A.V.S.H. Additionally, M.T.T.-L. was supported by CONACyT grant 14495; M.S. was supported by the Chicago Fellows program of the University of Chicago and PAPIIT grant IA206122 of the National Autonomous University of Mexico. Training activities in Mexico were hosted by CINVESTAV and supported in part by

CABANA, a capacity-strengthening project for bioinformatics in Latin America, financially supported by the Global Challenges Research Fund of the UK (RCUK/BB/P027849/1).

**Author contributions** Study conception, design and funding: A.M.-E., L.G.-G., S.L.F.-V., A.J.M. and A.V.S.H. Genotyping and data quality control: C.D.Q.-C., A.Y.C., M.J.P.-M. and M.S. Phenotypic dataset generation and curation: L.P.C.-H., L.F.-R., G.D.-S., N.M.-R., S.C.-Q., E.F.-G. and A.Y.C. Biochemical measurements: M.T.T.-L., H.M.-M., C.A.A.-S. and L.F.-R. Groupings and classification: M.S., V.A.-A., J.N. and A.M.-E. Population genetics: M.S., M.J.P.-M., A.Y.C., C.B.-J., S.G.M.-M., A.R., C.R.G., G.L.W., J.N. and A.M.-E. Genotypic imputation: A.J.-K., A.Y.C., A.C., K.A., A.J.M. and A.V.S.H. GWAS: A.Y.C., A.J.M., K.A., A.C. and A.V.S.H. Complex trait modelling: M.S., G.L.W., J.N., A.M.-E. and L.G.-G. Manuscript writing: M.S., with input and edits from A.Y.C., G.D.-S., A.G.I., S.L.F.-V., M.T.T.-L., A.J.M., J.N., L.G.-G. and A.M.-E. Project coordination: A.M.-E. and L.G.-G.

**Competing interests** The authors declare no competing interests.

**Additional information**
**Correspondence and requests for materials** should be addressed to Mashaal Sohail, Alexander J. Mentzer, Lourdes García-García or Andrés Moreno-Estrada.

**A**

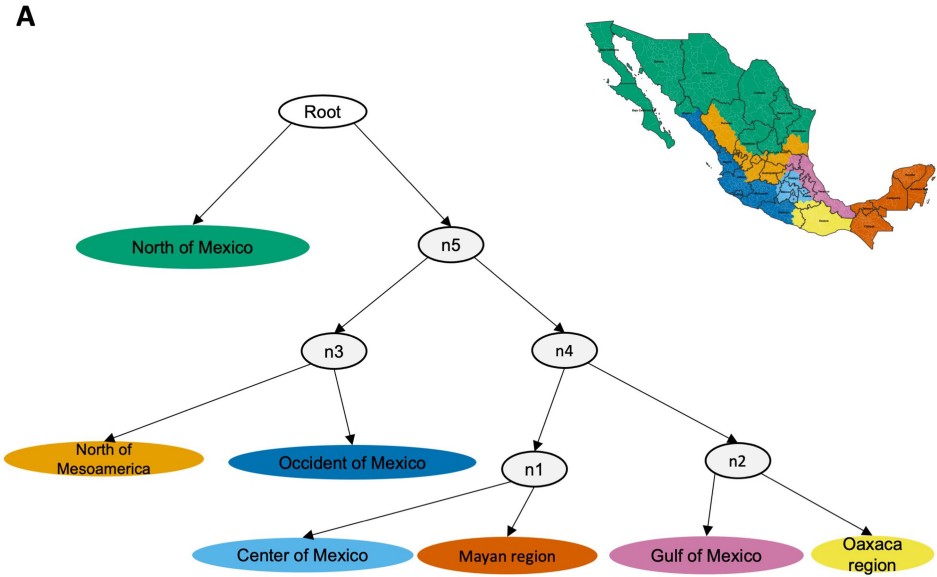

**B**

| | MXB-GWAS based | | | | UKB-GWAS based | | | |
|---|---|---|---|---|---|---|---|---|
| | Imputed dataset | | Genotyped dataset | | Imputed dataset | | Genotyped dataset | |
| | $R^2$ | P | $R^2$ | P | $R^2$ | P | $R^2$ | P |
| Height | 0.017 | 0.1 | 0.017 | 0.1 | -0.021 | $1\times10^{-8}$ | 0.053 | $1\times10^{-5}$ |
| BMI | 0.006 | 0.1 | 0.006 | 0.1 | 0.004 | $1\times10^{-8}$ | 0.016 | $1\times10^{-5}$ |
| Triglycerides | 0.045 | $1\times10^{-8}$ | 0.042 | $1\times10^{-8}$ | 0.039 | $1\times10^{-8}$ | 0.051 | $1\times10^{-5}$ |
| Cholesterol | 0.023 | 0.1 | 0.012 | 0.1 | 0.023 | $1\times10^{-8}$ | 0.023 | $1\times10^{-8}$ |
| HDL | 0.031 | $1\times10^{-8}$ | 0.026 | $1\times10^{-8}$ | 0.017 | $1\times10^{-8}$ | 0.034 | $1\times10^{-8}$ |
| LDL | 0.009 | $1\times10^{-8}$ | 0.016 | $1\times10^{-8}$ | 0.012 | $1\times10^{-8}$ | 0.02 | $1\times10^{-5}$ |
| Glucose | 0.027 | 0.1 | 0.018 | 0.1 | -0.003 | 0.01 | 0.014 | $1\times10^{-5}$ |
| Creatinine | 0.049 | 0.1 | 0.035 | 0.1 | 0.019 | $1\times10^{-8}$ | 0.034 | 0.001 |
| Systolic blood pressure | 0.003 | 0.1 | 0.004 | $1\times10^{-5}$ | -0.003 | $1\times10^{-8}$ | 0.012 | $1\times10^{-8}$ |
| Diastolic blood pressure | $-1.7\times10^{-5}$ | $1\times10^{-8}$ | 0.003 | 0.1 | -0.005 | $1\times10^{-8}$ | 0.003 | $1\times10^{-8}$ |

**Extended Data Fig. 1 | Genetic histories and polygenic prediction in the MXB.**
A) Admixture histories of individuals in different cultural regions using an AdmixtureBayes approach. Here the admixture graph with the highest posterior probability is shown, inferred using genomic regions with ancestries from the Americas. Internal inferred ancestral node populations are colored grey. The tree is rooted using the Han as an outgroup. B) Trait variance explained and p-value threshold of best predictive polygenic score using MXB-GWAS-based or UKB-GWAS-based prediction. Polygenic scores were computed using SNPs significant at five different p-value thresholds (0.1, 0.01, 0.001, 0.00001, $10^{-8}$). A linear null model was created for each trait including age, sex and 10 principal components as covariates. A second polygenic score model was created adding the polygenic score to the null model. We computed the $R^2$ of the polygenic score by taking the difference between the $R^2$ of the polygenic score model and the $R^2$ of the null model. The maximum $R^2$ was used to the pick the p-value threshold for the best predictive polygenic score shown in the table.

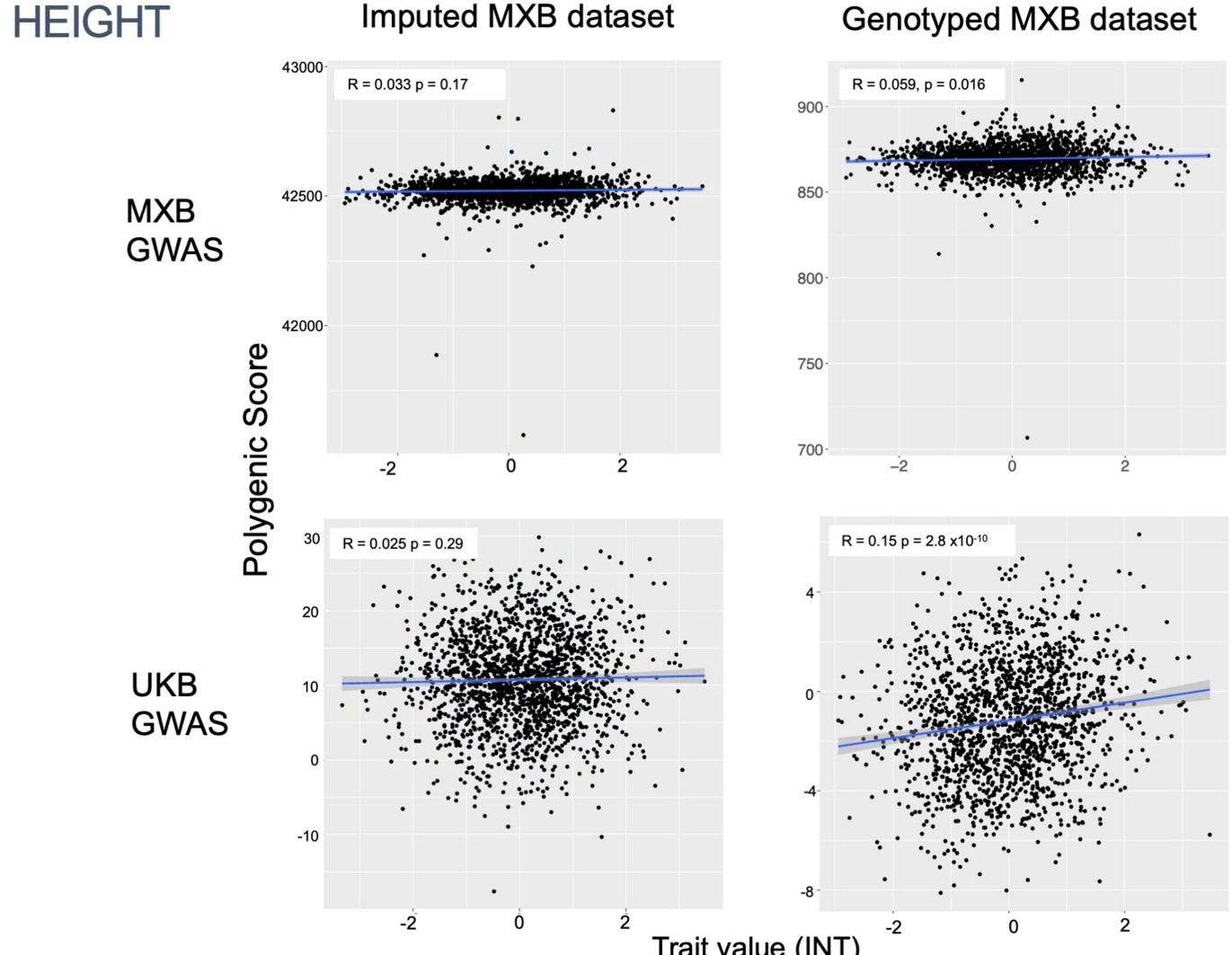

**Extended Data Fig. 2 | Prediction performance of MXB-GWAS-based or UKB-GWAS-based polygenic scores computed for height in the MXB.** Traits are inverse normalized. Prediction performance is measured by the correlation between polygenic score (the sum of all alleles associated at p < 0.1 weighted by their estimated effect sizes) and trait value (Pearson correlation R and associated two-sided p-value). Smoothed conditional mean lines are shown using a linear model. Error bands represent 95% confidence intervals.

## BMI

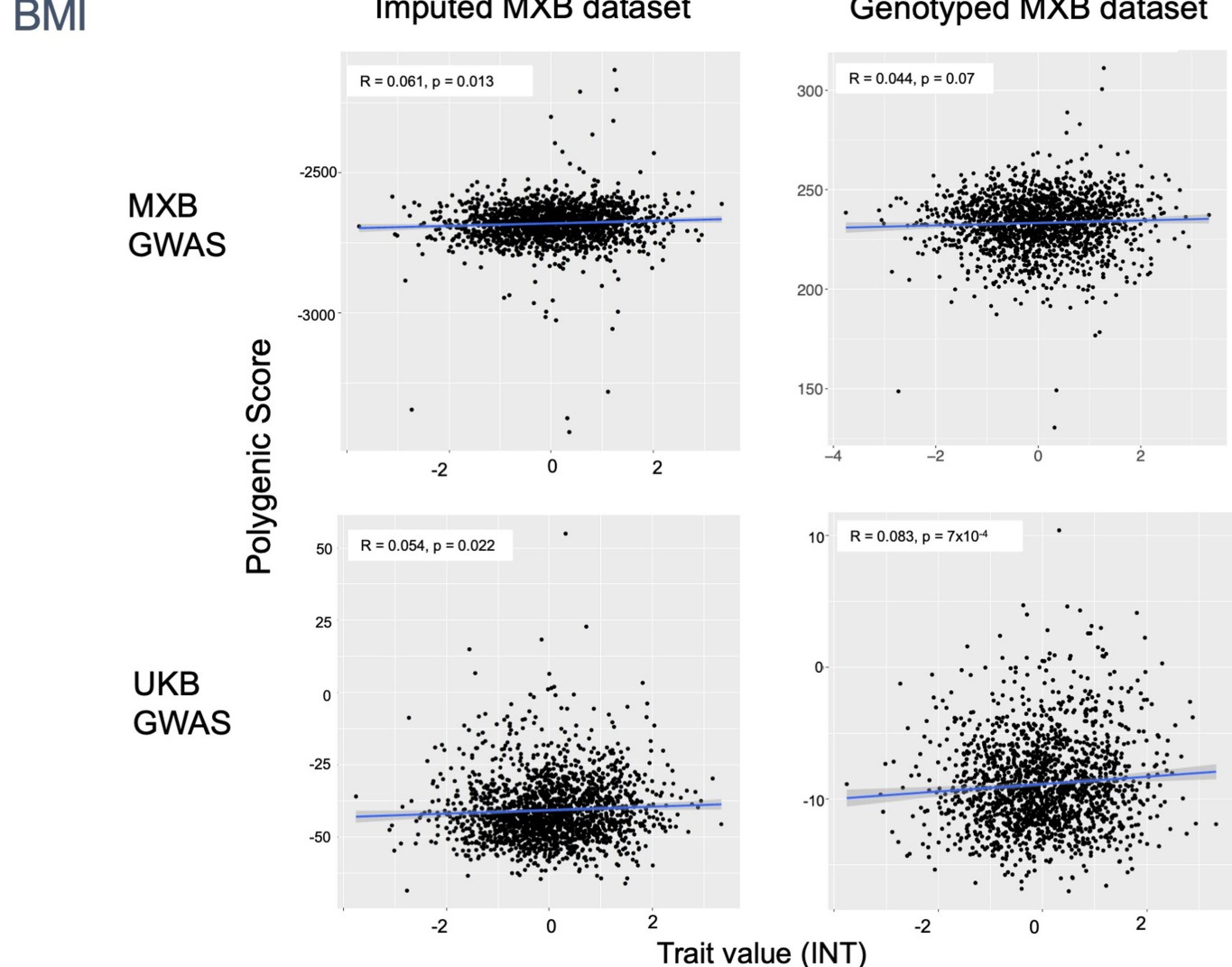

**Extended Data Fig. 3 | Prediction performance of MXB-GWAS-based or UKB-GWAS-based polygenic scores computed for BMI in the MXB.** Traits are inverse normalized. Prediction performance is measured by the correlation between polygenic score (the sum of all alleles associated at p < 0.1 weighted by their estimated effect sizes) and trait value (Pearson correlation R and associated two-sided p-value). Smoothed conditional mean lines are shown using a linear model. Error bands represent 95% confidence intervals.

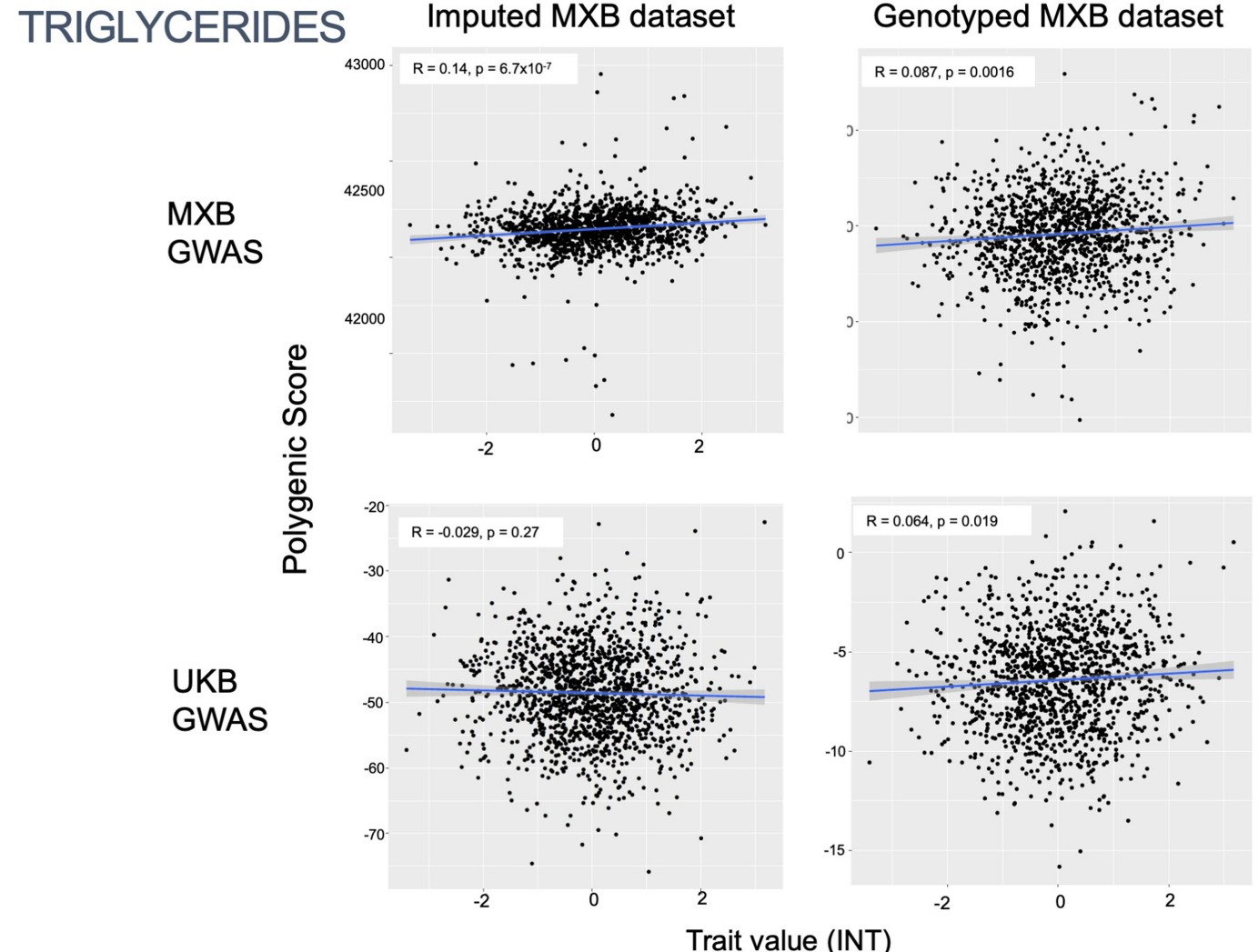

**Extended Data Fig. 4 | Prediction performance of MXB-GWAS-based or UKB-GWAS-based polygenic scores computed for triglycerides in the MXB.** Traits are inverse normalized. Prediction performance is measured by the correlation between polygenic score (the sum of all alleles associated at p < 0.1 weighted by their estimated effect sizes) and trait value (Pearson correlation R and associated two-sided p-value). Smoothed conditional mean lines are shown using a linear model. Error bands represent 95% confidence intervals.

## CHOLESTEROL

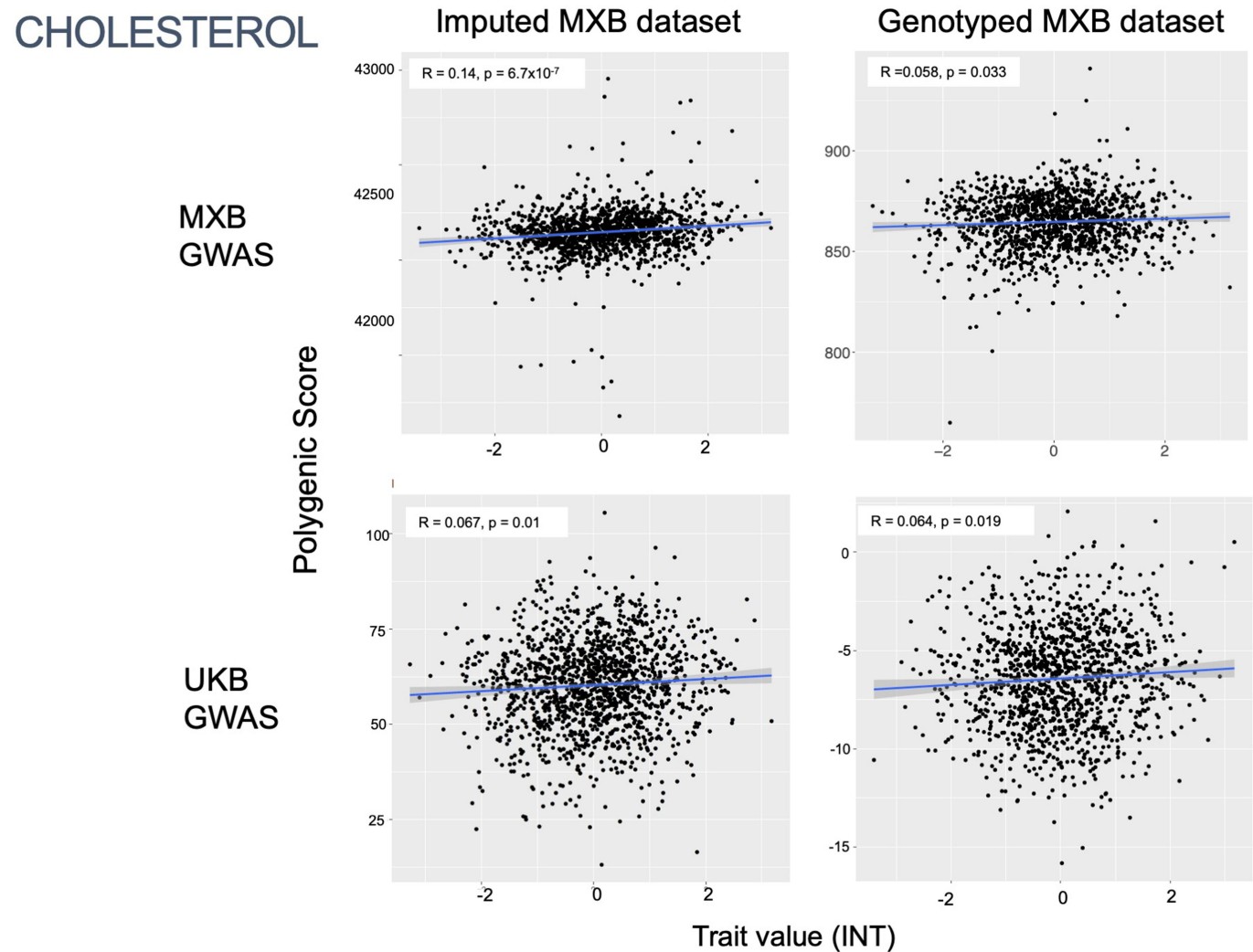

**Extended Data Fig. 5 | Prediction performance of MXB-GWAS-based or UKB-GWAS-based polygenic scores computed for cholesterol in the MXB.** Traits are inverse normalized. Prediction performance is measured by the correlation between polygenic score (the sum of all alleles associated at p < 0.1 weighted by their estimated effect sizes) and trait value (Pearson correlation R and associated two-sided p-value). Smoothed conditional mean lines are shown using a linear model. Error bands represent 95% confidence intervals.

# HDL

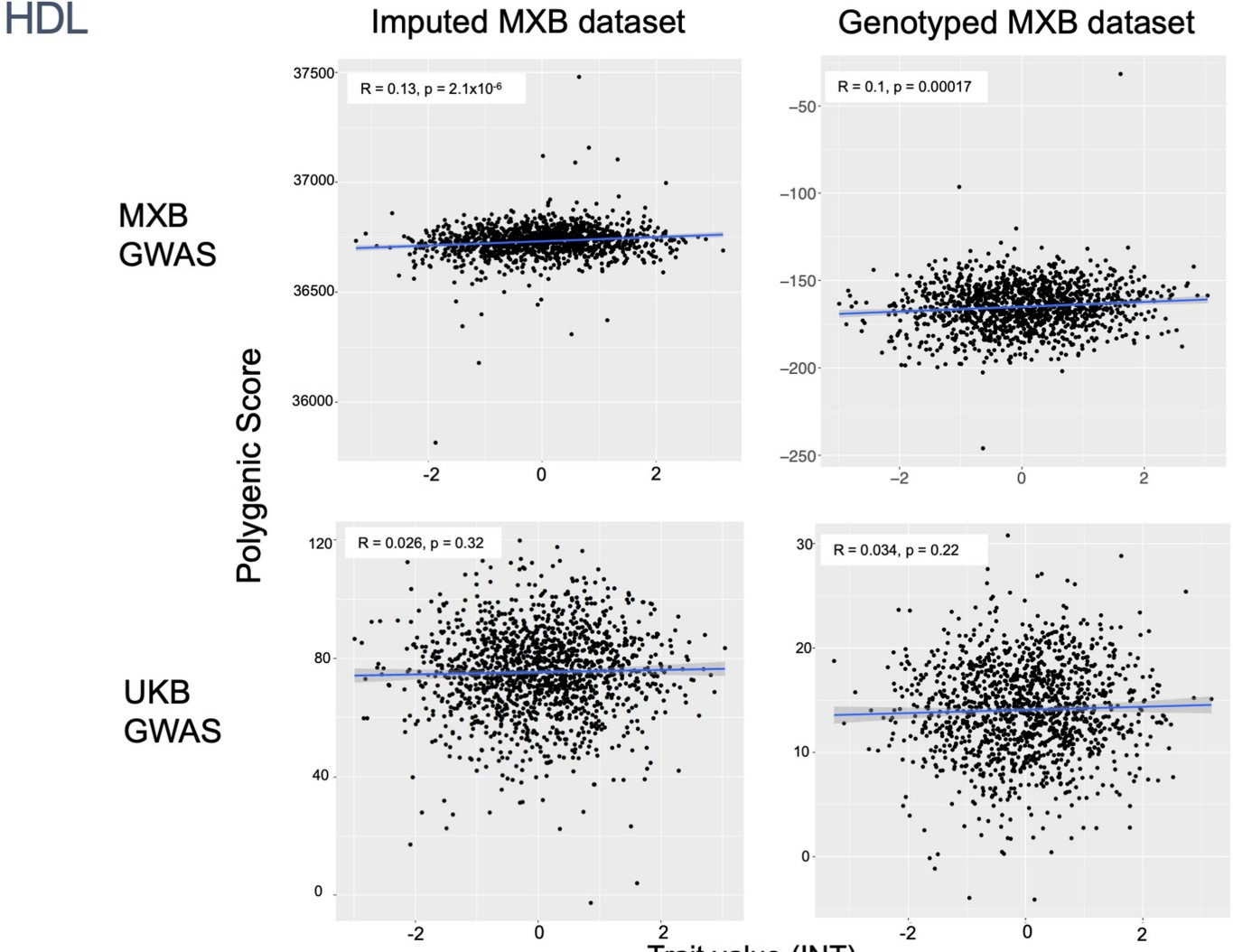

**Extended Data Fig. 6 | Prediction performance of MXB-GWAS-based or UKB-GWAS-based polygenic scores computed for HDL in the MXB.** Traits are inverse normalized. Prediction performance is measured by the correlation between polygenic score (the sum of all alleles associated at p < 0.1 weighted by their estimated effect sizes) and trait value (Pearson correlation R and associated two-sided p-value). Smoothed conditional mean lines are shown using a linear model. Error bands represent 95% confidence intervals.

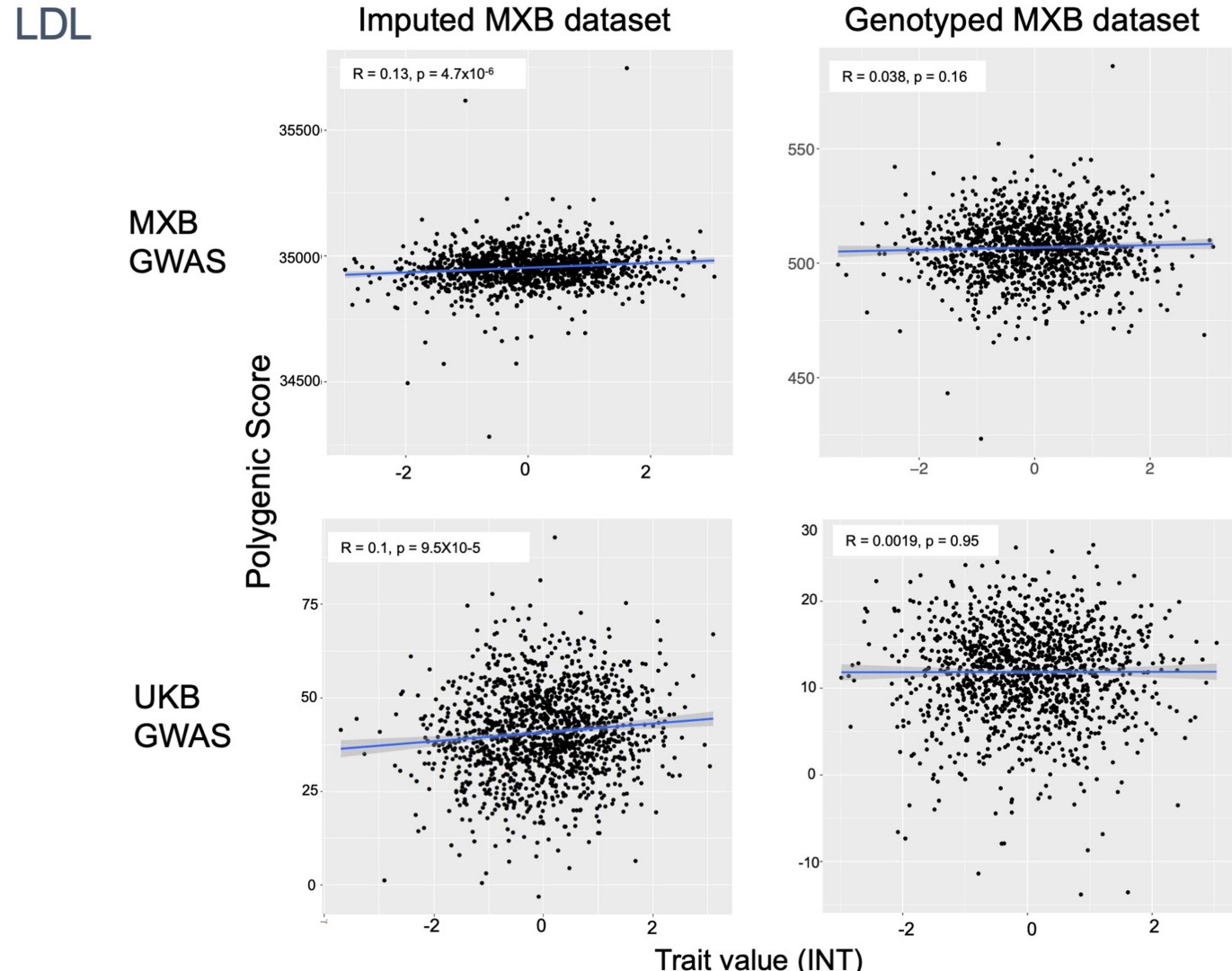

**Extended Data Fig. 7 | Prediction performance of MXB-GWAS-based or UKB-GWAS-based polygenic scores computed for LDL in the MXB.** Traits are inverse normalized. Prediction performance is measured by the correlation between polygenic score (the sum of all alleles associated at p < 0.1 weighted by their estimated effect sizes) and trait value (Pearson correlation R and associated two-sided p-value). Smoothed conditional mean lines are shown using a linear model. Error bands represent 95% confidence intervals.

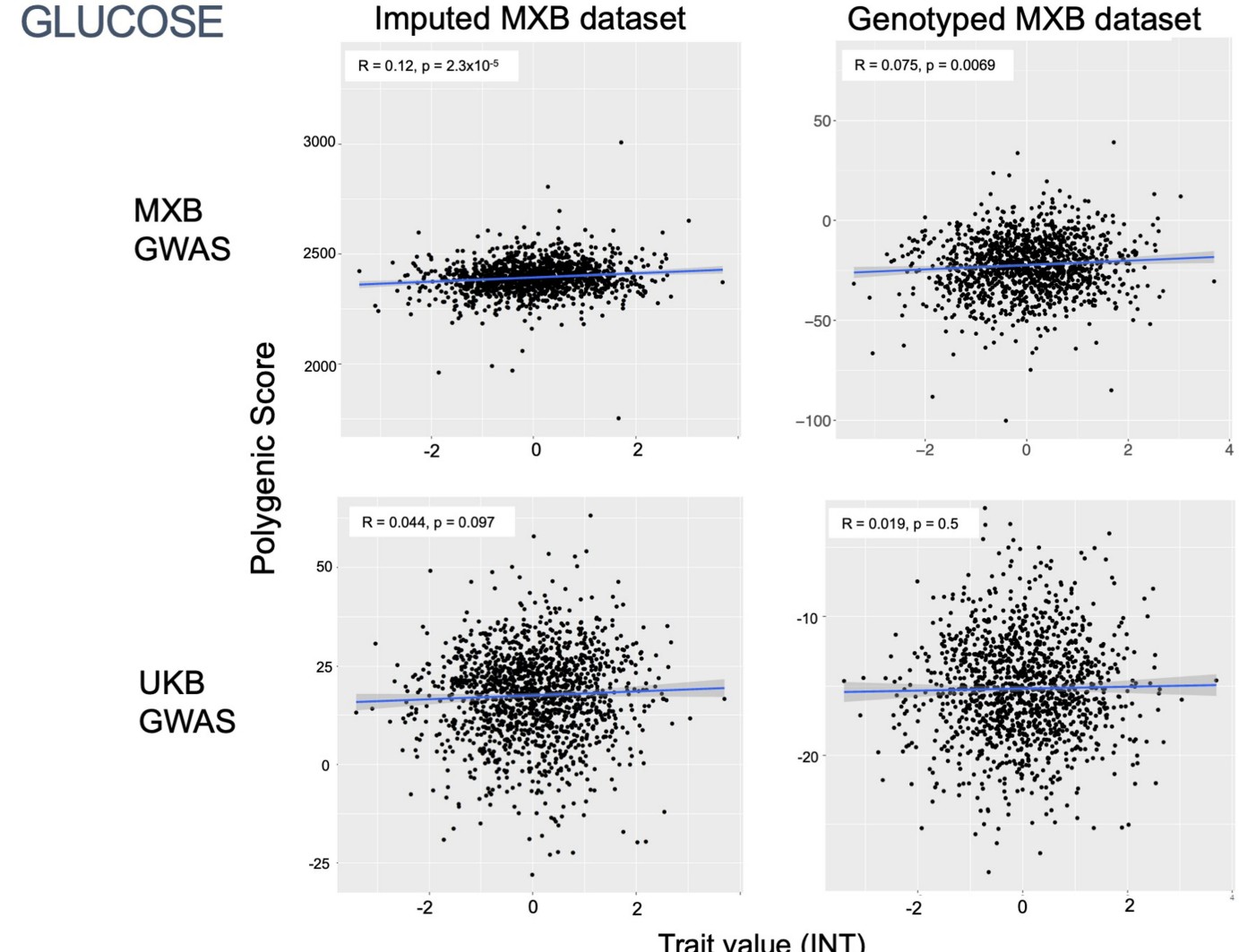

**Extended Data Fig. 8 | Prediction performance of MXB-GWAS-based or UKB-GWAS-based polygenic scores computed for glucose in the MXB.** Traits are inverse normalized. Prediction performance is measured by the correlation between polygenic score (the sum of all alleles associated at p < 0.1 weighted by their estimated effect sizes) and trait value (Pearson correlation R and associated two-sided p-value). Smoothed conditional mean lines are shown using a linear model. Error bands represent 95% confidence intervals.

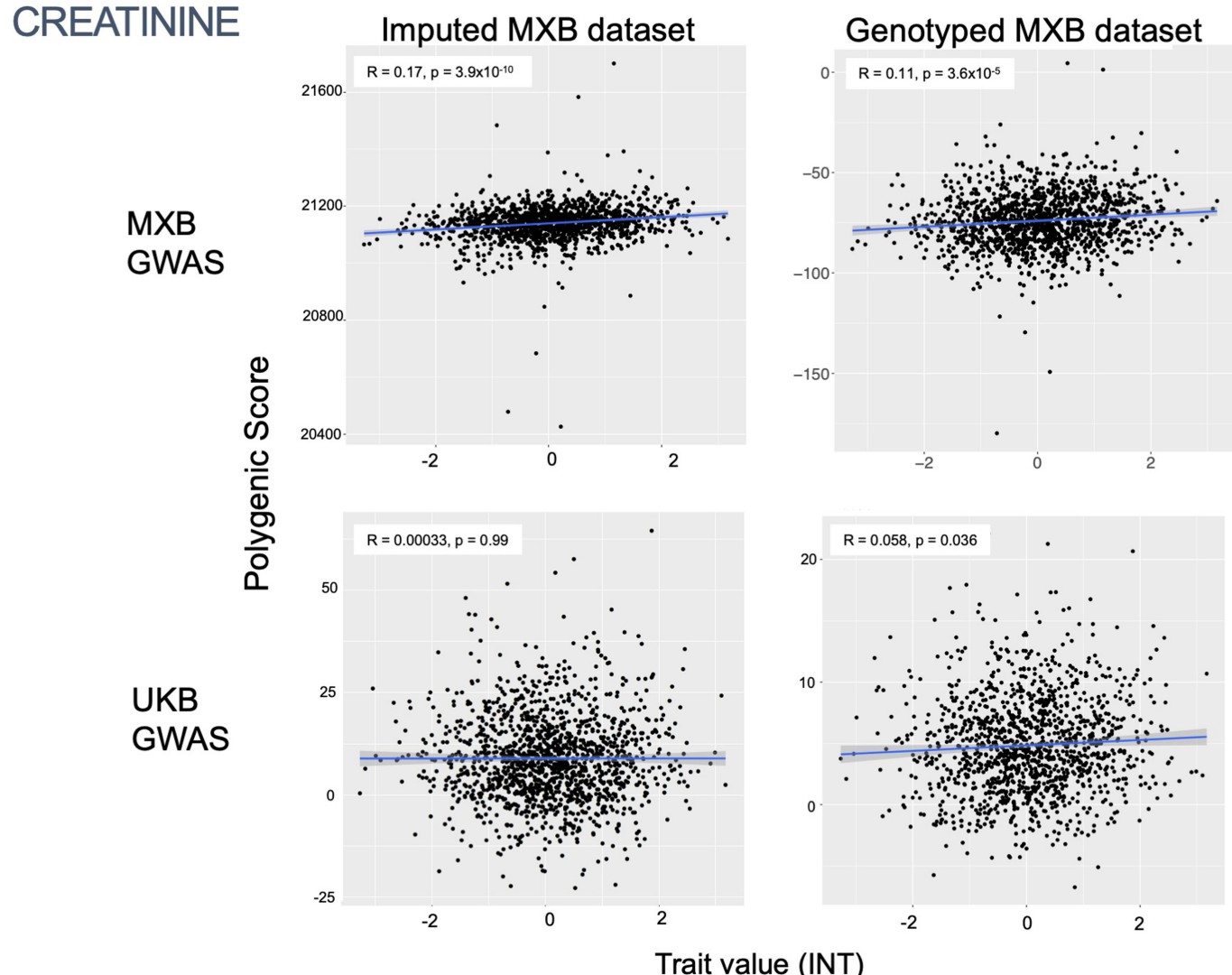

**Extended Data Fig. 9 | Prediction performance of MXB-GWAS-based or UKB-GWAS-based polygenic scores computed for creatinine in the MXB.** Traits are inverse normalized. Prediction performance is measured by the correlation between polygenic score (the sum of all alleles associated at p < 0.1 weighted by their estimated effect sizes) and trait value (Pearson correlation R and associated two-sided p-value). Smoothed conditional mean lines are shown using a linear model. Error bands represent 95% confidence intervals.

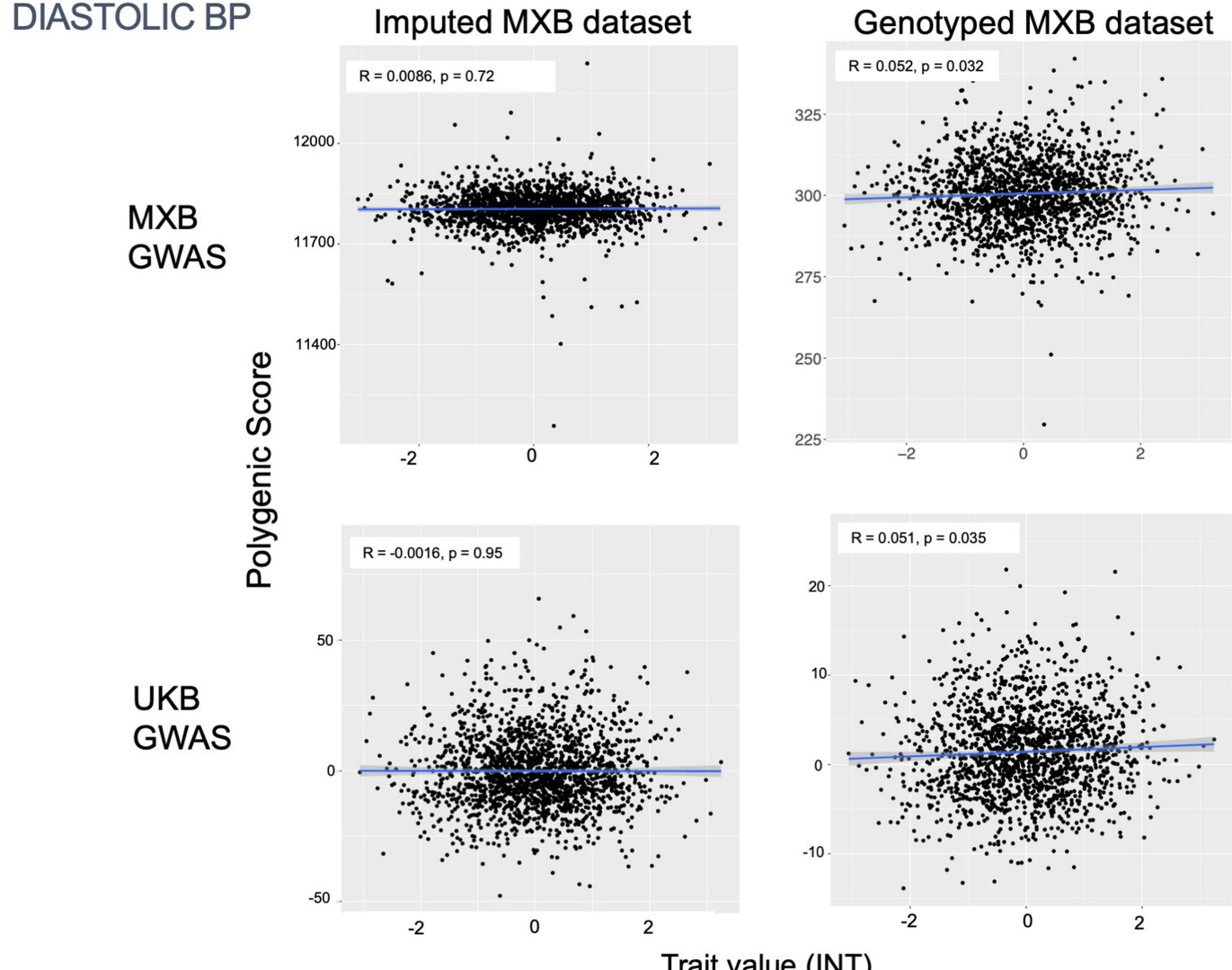

**Extended Data Fig. 10 | Prediction performance of MXB-GWAS-based or UKB-GWAS-based polygenic scores computed for diastolic blood pressure in the MXB.** Traits are inverse normalized. Prediction performance is measured by the correlation between polygenic score (the sum of all alleles associated at p < 0.1 weighted by their estimated effect sizes) and trait value (Pearson correlation R and associated two-sided p-value). Smoothed conditional mean lines are shown using a linear model. Error bands represent 95% confidence intervals.

# Reporting Summary

## Statistics

For all statistical analyses, confirm that the following items are present in the figure legend, table legend, main text, or Methods section.

| n/a | Confirmed | |
|---|---|---|
| ☐ | ☒ | The exact sample size (*n*) for each experimental group/condition, given as a discrete number and unit of measurement |
| ☐ | ☒ | A statement on whether measurements were taken from distinct samples or whether the same sample was measured repeatedly |
| ☐ | ☒ | The statistical test(s) used AND whether they are one- or two-sided *Only common tests should be described solely by name; describe more complex techniques in the Methods section.* |
| ☐ | ☒ | A description of all covariates tested |
| ☐ | ☒ | A description of any assumptions or corrections, such as tests of normality and adjustment for multiple comparisons |
| ☐ | ☒ | A full description of the statistical parameters including central tendency (e.g. means) or other basic estimates (e.g. regression coefficient) AND variation (e.g. standard deviation) or associated estimates of uncertainty (e.g. confidence intervals) |
| ☐ | ☒ | For null hypothesis testing, the test statistic (e.g. *F*, *t*, *r*) with confidence intervals, effect sizes, degrees of freedom and *P* value noted *Give P values as exact values whenever suitable.* |
| ☐ | ☒ | For Bayesian analysis, information on the choice of priors and Markov chain Monte Carlo settings |
| ☐ | ☒ | For hierarchical and complex designs, identification of the appropriate level for tests and full reporting of outcomes |
| ☐ | ☒ | Estimates of effect sizes (e.g. Cohen's *d*, Pearson's *r*), indicating how they were calculated |

*Our web collection on statistics for biologists contains articles on many of the points above.*

## Software and code

Policy information about availability of computer code

**Data collection**
Our samples were genotyped on the Illumina's Multi-Ethnic Genotyping Array (MEGA). The design of this array was previously led by Christopher Gignoux and Genevieve Wojcik, co-authors on this manuscript. Several properties place the MEGAex array as the ideal choice for biobank genotyping. It captures 1,748,250 SNPs derived from admixed population studies, making it broadly applicable in diverse populations. Genome Studio was used to convert raw image files to plink files with raw genotype information. All SNPs were flipped to the forward strand, and duplicate SNPs were removed. For sites with missing chromosome number, physical position or both, we updated the map using the information in the SNP name or by mapping their rsID using dbSNP Build 151.
We use plink to remove all individuals with more than 5% missing genotype data, and all genotypes with more than 5% missing individuals. We restricted the analyses to only autosomes and removed all monomorphic SNPs. We restricted the analysis to only biallelic SNPs and removed all SNPs with ambiguous strand for all downstream analyses using SNPFLIP. All related individuals were detected using plink (--Z-genome --min 0.5) after pruning for LD (--indep-pairwise 50 5 0.5). A script was written to iteratively find nodes and remove related individuals to obtain the final QC-ed dataset.

**Data analysis**
EIGENSOFT (v7.2.1): https://github.com/dReichLab/EIG
Smartpca (part of Eigensoft v7.2.1): https://github.com/chrchang/eigensoft/tree/master/POPGEN
ADMIXTURE (v1.3.0): https://dalexander.github.io/admixture/
UMAP (repository downloaded Dec 2021): https://github.com/lmcinnes/umap
Archetypal analysis (repository downloaded Nov 2022): https://github.com/AI-sandbox/archetypal-analysis
GNOMIX (repository downloaded Oct 2021): https://github.com/AI-sandbox/gnomix
maas-MDS (repository downloaded Nov 2021): https://github.com/AI-sandbox/maasMDS/
shapeit (v2.17): https://mathgen.stats.ox.ac.uk/genetics_software/shapeit/shapeit.html
RFMIX (v2): https://github.com/slowkoni/rfmix

PCAmask (20131203): https://mybiosoftware.com/tag/pcamask
AdmixtureBayes (repository downloaded Jan 2023): https://github.com/svendvn/AdmixtureBayes
Refined-ibd (17Jan20): https://faculty.washington.edu/sguy/asibdne/
Merge-ibd-segments (17Jan20): https://faculty.washington.edu/sguy/asibdne/
Beagle (25Nov19): https://faculty.washington.edu/sguy/asibdne/
asIBDNe (19Sept19): https://faculty.washington.edu/sguy/asibdne/
Plink (v1.9): https://www.cog-genomics.org/plink/1.9/
Variant Effect Predictor (ensemble-vep-release-104): https://www.ensembl.org/info/docs/tools/vep/index.html
Regenie (v3.1.3): https://rgcgithub.github.io/regenie/
FINEMAP (v1.3): http://www.christianbenner.com/
FUMA (v1.4.1): https://github.com/Kyoko-wtnb/FUMA-webapp/
KING (v2.2.8): https://www.kingrelatedness.com/
Mxmaps (2020.1.1.9000): https://www.diegovalle.net/mxmaps/
Genesis (Release 3.17): https://www.bioconductor.org/packages/release/bioc/html/GENESIS.html
lme4qtl (development version): https://github.com/variani/lme4qtl

For manuscripts utilizing custom algorithms or software that are central to the research but not yet described in published literature, software must be made available to editors and reviewers. We strongly encourage code deposition in a community repository (e.g. GitHub). See the Nature Portfolio guidelines for submitting code & software for further information.

# Data

Policy information about availability of data

All manuscripts must include a data availability statement. This statement should provide the following information, where applicable:
- Accession codes, unique identifiers, or web links for publicly available datasets
- A description of any restrictions on data availability
- For clinical datasets or third party data, please ensure that the statement adheres to our policy

The dataset for the 6,057 newly genotyped individuals from the MX biobank project are available at the European Genome-phenome Archive (EGA) through a Data Access Agreement with the Data Access Committee (EGA accession number for study: EGAS00001005797; Datasets: EGAD00010002361 "Mexican_Biobank_Genotypes" and EGAD00001008354 "Mexican Biobank 50 Genomes"). GWAS summary statistics generated as part of this study are available at: https://doi.org/10.5281/zenodo.7420254. Variant Effect Predictor (VEP, https://asia.ensembl.org/info/docs/tools/vep/index.html) was used to annotate the effect of a variant using the humdiv database. 1000 Genomes data was accessed from: http://ftp.1000genomes.ebi.ac.uk/vol1/ftp/release/20130502/. HGDP data was downloaded from: https://rosenberglab.stanford.edu/hgdpsnpDownload.html. Pan-ancestry GWAS summarty statistics from UK Biobank were downloaded from: https://pan.ukbb.broadinstitute.org/docs/summary.

# Human research participants

Policy information about studies involving human research participants and Sex and Gender in Research.

| Reporting on sex and gender | Self-reported sex was collected as part of the survey, and sex was also identified using genetic plink analysis. There are self-reported 1844 males and 4213 females in the entire dataset. There were a few cases (249) where plink reports an ambiguous sex. For our complex trait analyses, we used self-reported sex as a covariate. Prior to GWAS and polygenic score analyses, individuals with mismatched sex or IBD>0.9 were removed. |
|---|---|
| Population characteristics | Trained personnel conducted the interviews. Information was collected on household and sociodemographic characteristics, current health status, health care service usage, and behavioral aspects of participants. The genotyped and QC'ed data is 30.45% Male and 69.55% Female. The age distribution is as follows: 20-39 (55.13%), 40-59 (30.35%), 60-79 (12.77%) and 80+ (1.86%). The dataset is ~70% from rural areas and ~30% from urban areas. |
| Recruitment | Since 1988, Mexico has established periodical health surveys (ENSA) for surveillance of Mexican population-based health and nutrition metrics. In this study, we use data and samples collected by the National Institute of Public Health (INSP) from the survey performed in 2000, the ENSA 2000. This survey was a probabilistic, multi-stage, stratified, cluster household survey conducted by the Mexican Secretariat of Health from November 1999 to June 2000. Research design and methods have been described elsewhere. Participants were randomly selected in order to be representative of the civilian, non-institutionalized Mexican population, at the state and national levels. |
| Ethics oversight | The ENSA 2000 was carried out following the strictest ethical principles and in accordance with the Helsinki Declaration of Human Studies. Informed consent was obtained from all participants. Extracted DNA has been stored and maintained at the National Institute of Public Health (Cuernavaca, Mexico), and samples were genotyped and analyzed at the Advanced Genomics Unit of CINVESTAV (Irapuato, Mexico) through a collaborative institutional agreement. The project was reviewed and approved by the Research Ethics Committee and the Biosafety Committee of the National Institute of Public Health (IRB approvals CI: 1479 and CB: 1470). For the present project, personally identifiable data was removed from the data set. |

Note that full information on the approval of the study protocol must also be provided in the manuscript.

# Field-specific reporting

Please select the one below that is the best fit for your research. If you are not sure, read the appropriate sections before making your selection.

☒ Life sciences  ☐ Behavioural & social sciences  ☐ Ecological, evolutionary & environmental sciences

For a reference copy of the document with all sections, see nature.com/documents/nr-reporting-summary-flat.pdf

# Life sciences study design

All studies must disclose on these points even when the disclosure is negative.

| | |
|---|---|
| Sample size | No statistical methods were used to predetermine sample size. The sample size was selected before analysis was begun based on available samples and budgetary constraints for genotyping. We sought to include sufficient sample to power statistical comparisons. |
| Data exclusions | We use plink to remove all individuals with more than 5% missing genotype data, and all genotypes with more than 5% missing individuals. We restricted the analyses to only autosomes and removed all monomorphic SNPs. We restricted the analysis to only biallelic SNPs and removed all SNPs with ambiguous strand for all downstream analyses using SNPFLIP. All related individuals were detected using plink (--Z-genome --min 0.5) after pruning for LD (--indep-pairwise 50 5 0.5). A script was written to iteratively find nodes and remove related individuals to obtain the final QC-ed dataset. Prior to GWAS and polygenic score analyses, individuals with mismatched sex or IBD>0.9 were removed. |
| Replication | No experiments were conducted, so replication of experimental results is not relevant. |
| Randomization | For some analyses where covariates were relevant such as the complex trait analyses, they were controlled for in a mixed model framework using lme4qtl. For the GWAS analysis, individuals were either allocated into cases and controls groups for binary traits or analyzed in a single group for quantitative traits. In both cases, covariates such as the genetic relationship matrix, principal components, age and sex were controlled for using a novel machine-learning method called REGENIE for fitting a whole-genome regression model. For all other analyses, either all individuals were used, or a subset of individuals with high proportion of ancestries from the Americas (as inferred using Admixture) were used when we wanted to focus on patterns in specifically the indigenous genetic background. |
| Blinding | Blinding was not relevant to this study, as it is not a clinical trial but rather a descriptive population genetics analysis and association study. The individuals were not given a drug for which the effects were evaluated, thus making blinding irrelevant. |

# Reporting for specific materials, systems and methods

We require information from authors about some types of materials, experimental systems and methods used in many studies. Here, indicate whether each material, system or method listed is relevant to your study. If you are not sure if a list item applies to your research, read the appropriate section before selecting a response.

## Materials & experimental systems

| n/a | Involved in the study |
|---|---|
| ☒ ☐ | Antibodies |
| ☒ ☐ | Eukaryotic cell lines |
| ☒ ☐ | Palaeontology and archaeology |
| ☒ ☐ | Animals and other organisms |
| ☒ ☐ | Clinical data |
| ☒ ☐ | Dual use research of concern |

## Methods

| n/a | Involved in the study |
|---|---|
| ☒ ☐ | ChIP-seq |
| ☒ ☐ | Flow cytometry |
| ☒ ☐ | MRI-based neuroimaging |

