## [Peer Review File · Nature]

Manuscript Title: Mexican Biobank advances population and medical genomics of diverse ancestries

Reviewer Comments & Author Rebuttals

Reviewer Reports on the Initial Version:

Referees' comments:

Referee #1 (Remarks to the Author):

The manuscript describes a population genetics study based on the Mexico Biobank (MXB) with a study design that aimed to maximize geographic coverage of Mexico and representation of participants who speak an indigenous language and live in a rural area. Participants were genotyped on the Illumina Multi-Ethnic Genotyping Array and had a set of linked phenotypes available. Analyses of genetic structure and admixture yielded a fine-grained resolution of ancestries and migrations which were linked to contextual archaeological and historical data. Ancestry-specific IBD analyses were used to resolve genetic history within Mexico of various lineages, ranging from indigenous pre-colonization lineages to other lineages introduced from European, Asian and African migrations. Analyses of runs-of-homozygosity (ROH) showed that the prevalence of small ROH was correlated with ancestry proxies and this was probably a result of historical bottlenecks and small population sizes. Rare deleterious variant burden was found to be correlated with ancestry proxies, with positive correlation with ancestries from Western Europe and Africa and negative correlation with ancestries from Central America. The modeling of the contribution of genetic and environmental variation to complex traits revealed that Central American ancestries had a significant association with variation in height, fasting glucose and triglyceride levels while ROH was associated with triglycerides and body mass index (BMI).

The study design is reasonable, given the study objectives. The study methods and analyses are well-described. Most of the findings related to population structure, admixture and ROH recapitulate the findings of previous studies conducted in Mexico or as part of larger studies (such as the 1000 Genomes Project). However, the fine-grained resolution of ancestry and admixture with the integration of archaeological and historical information is an important advance. Strengths of the study include the large sample size, representativeness of the study population, integration with archaeological and anthropological evidence, and the complex trait modeling in the context of genetic and environmental factors.

Major concerns are:

1. The models of the complex traits did not include an important measure of genomewide risk, namely the polygenic risk score (PRS). Each of the traits should be re-analyzed to include its PRS as one of the covariates to properly evaluate the contribution of genomewide risk factors currently included in the models.

2. There is an important key covariate (i.e. BMI) which should be included in the analysis of traits such as fasting glucose, triglycerides, HDL-cholesterol, LDL-cholesterol, systolic blood pressure and diastolic blood pressure. For these traits, age, sex and BMI are the three covariates with the largest effect sizes in most populations. The authors included age and sex, but not BMI, in their models. The analyses should be done with the inclusion of BMI. Otherwise, the presented effect sizes of the factors and model r^2 could be considered unreliable.

3. The analysis of the genomic burden of rare deleterious mutations (and of other classes of variants) is based on array genotypes. Most arrays have a SNP ascertainment bias as well as design biases (based on the objectives of the array), and may not adequately mirror whole genome sequences (WGS). While this point is acknowledged in part in the manuscript, the current comparison with the subset of Mexicans living in Los Angeles (1000 Genomes MXL) is insufficient and is based on the assumption that the 1000 Genomes MXL are representative of most or all of the groups in the current study. The authors should discuss further how likely their findings will remain consistent if they had whole genome sequence (WGS) in the MXB participants that they studied.

Referee #2 (Remarks to the Author):

The authors based their manuscript on an interesting dataset for medical and population studies, including more than 6,000 individuals from different regions of Mexico. The manuscript covers many topics to infer the relationship between regional demography, ancestry, genetic load, environment, and health data. However, the different results are not well connected throughout the text. Although the database is quite interesting, the reported findings' novelty is unclear, all of which were briefly described and hardly discussed concerning the literature and previous results. In many moments, the text seems only to corroborate previous studies; that is, more data were used to conclude what had already been published about the Mexican population.

The sample is the key point of this manuscript. Still, unfortunately, because it is not complete genomes, it is not something new or unpublished to Mexican populations, as there are several GWAS studies using large samples already published in this population (Spear et al. 2020; Martinez-Magana et al, 2021; Feofanova et al, 2020).

Specifically:

1. Regarding the ancestry analyses, I have doubts about what is really new in this study. For example, Moreno-Estrada had already demonstrated in his 2015 paper, also using array data from more than 1000 mestizos and native individuals, that the native ancestry of mestizos recapitulated pre-Columbian regional origin in Mexico. The present manuscript describes the same, only including more samples from different regions.

2. The authors describe the finding of East Asian ancestry related to the Manila galleon in Mexico. It

is surprising, given that the same authors have an article entirely dedicated to the subject (Rodriguez-Rodriguez et al. 2022), quite complete (not preliminary as described in the text), and recently published.

3. The most exciting part of the ancestry analysis is related to the demographic expansions in the different periods of the pre-contact history of Mexico. However, the results are presented in a very descriptive way, without significant discussions.

4. The methodology is quite simple; the authors could have used fine-structure and f-statistics methods to answer more profound questions about the origin and dynamics of the sample studied. The methodology used is also not new.

5. The ROH analyses seem really interesting, however, there seems to be some confusion regarding the theoretical part and subsequent interpretation of data. Mainly regards the inference of time and number of ROH. In general, this section is not clear.

For example, the following sentence seems odd:

A bottleneck event or a long-term small population size will result in a large number of small ROH.

The correct would be to relate the short ROH to ancient events. Usually, bottleneck or isolation creates specific haplotypes of a population. Short ROH measuring tens of kb probably reflects homozygosity for ancient haplotypes that contribute to local LD patterns. In other words, it would be necessary to have an extreme and ancient bottleneck (like those explained by the out-of-Africa). Or a population to have had its effective size reduced many generations ago (redundant with the bottleneck) to generate a large number of small ROHs. However, this doesn't need to occur over a long period in a small population. Also, for example, many native populations have expanded but still have many short ROHs.

In the following sentence:

Consanguinity or marriage between relatives would instead result in fewer but longer...

It will be better to reframe it as:

Consanguinity or marriage between close relatives would instead result in fewer but longer...

I strongly suggest the use of different sizes of fragments in the analysis (at least three classes) and consider small ROH only those smaller than 2Mb.

Still, concerning ROH, can the fact that young individuals have more Central American fragments be an effect of sample bias? How much of the population studied are young people? Is the age and region distribution homogeneous?

6. The association between genomic data, phenotype and environment is poorly explored, without a proper discussion of the results. Furthermore, none of the associated phenotypes is new or unheard

of for Mexican populations but the authors did not cite any previous articles published on the Mexican populations.

Referee #3 (Remarks to the Author):

In this manuscript, entitled 'Nationwide genomic biobank in Mexico unravels demographic history and complex trait architecture from 6,057 individuals', Sohail and colleagues reports genome-wide data from 6,057 individuals from Mexico. Through the genotyping of 1.8 million markers in all individuals, who come from 32 states in Mexico, they explore variation across Mexicans in ancestry proportions, IBD segments, and runs of homozygosity. Finally, they correlate ancestry and ROH with trait variation, focusing on some interesting traits such as height, BMI, and some other metabolism-related traits.

The manuscript is very well written, and I applaud all the aspect on diversity inclusion. Not only the genomics literature is strongly biased towards individuals of European ancestry but, in the case of Mexico, ancestry studies are particularly relevant because different ancestries are present, at varying extents, in the same populations. For me, this study presents a great resource for future diversity and genomic studies.

Having said that, I think that the study, in its current state, remains too preliminary and descriptive. Besides the larger sample size reported (which is great), most observations have already been published by the same authors (Rodriguez-Rodriguez et al. *Philos Trans R Soc Lond B Biol Sci* 2022; Ojeda-Granados et al. *Mol Biol Evol* 2022; Moreno-Estrada et al. *Science* 2014). This is the reason why I think the authors should consider the following major points, before proceeding further:

- Besides the descriptive analyses of PCA and admixture, I think a proper demographic inference would be a great asset for the paper, and I think the authors should perform proper demographic modeling with the data. In its current state, it remains just descriptive, and ones learns very little about the demographic history of the different ancestries observed in Mexico.
- The ancestry-specific analyses for East Asian and South Asian ancestries in Mexicans deserves further attention and supplementary analyses. Given that they are present in quite low frequencies in the populations, the authors should verify more formally their geographic origins and timing. For example, 1) how sensitive is the local ancestry inference to detect very low percentage ancestry tracts, e.g., here South Asian and East Asian tracts (moreover, South Asian is not super distant from European, and East Asian has also shared ancestry with American)? 2) given the low percentage, after masking other ancestries, the remaining South Asian/East Asian ancestry-specific SNPs must be very few and with lots of missing data, how confident of the analyses the authors are when dealing with such high missing data? To deal with both 1) and 2), I think the easiest way would be to run local ancestry inference on any published data that is known to have low percentage of East Asian/South Asian admixed ancestries; otherwise, simulation data would be helpful.
- The authors say that the identified correlation between rare variants and ancestries is not affected

by ascertainment bias but total mutation - ancestries correlation is. To me this contradicts the intuition of WGS vs. array. This should be supplemented with more formal analyses to show that the genotyping array used is free of ascertainment bias (see Fig S50).

- Importantly, although they correlate ancestry with variation in phenotypic traits, they do not report any GWAS data/statistics. I don't understand why they haven't gone further in these analyses. Given the sample size, I think, particularly for Nature, it is compulsory that they perform GWAS for each of the traits reported.

- The regression analyses are presented in a very descriptive way; they should be interpreted.

- The method to define IBD (IBDNe) has been recently criticized (e.g., <https://doi.org/10.1101/2022.08.03.501074>), so I think the authors should cross-check the validity of their results with an independent method.

Minor (but important)

The link to their custom scripts/approaches

(https://github.com/msohail88/MXB_popstruct_complextraits) is not working.

Author Rebuttals to Initial Comments:

Referees' comments:

Referee #1 (Remarks to the Author):

The manuscript describes a population genetics study based on the Mexico Biobank (MXB) with a study design that aimed to maximize geographic coverage of Mexico and representation of participants who speak an indigenous language and live in a rural area. Participants were genotyped on the Illumina Multi-Ethnic Genotyping Array and had a set of linked phenotypes available. Analyses of genetic structure and admixture yielded a fine-grained resolution of ancestries and migrations which were linked to contextual archaeological and historical data. Ancestry-specific IBD analyses were used to resolve genetic history within Mexico of various lineages, ranging from indigenous pre-colonization lineages to other lineages introduced from European, Asian and African migrations. Analyses of runs-of-homozygosity (ROH) showed that the prevalence of small ROH was correlated with ancestry proxies and this was probably a result of historical bottlenecks and small population sizes. Rare deleterious variant burden was found to be correlated with ancestry proxies, with positive correlation with ancestries from Western Europe and Africa and negative correlation with ancestries from Central America. The modeling of the contribution of genetic and environmental variation to complex traits revealed that Central American ancestries had a significant association with variation in height, fasting glucose and triglyceride levels while ROH was associated with triglycerides and body mass index (BMI).

The study design is reasonable, given the study objectives. The study methods and analyses are well-described. Most of the findings related to population structure, admixture and ROH recapitulate the findings of previous studies conducted in Mexico or as part of larger studies (such as the 1000 Genomes Project). However, the fine-grained resolution of ancestry and admixture with the integration of archaeological and historical information is an important advance. Strengths of the study include the large sample size, representativeness of the study population, integration with archaeological and anthropological evidence, and the complex trait modeling in the context of genetic and environmental factors.

We thank the reviewer for accurately summarizing our study and its strengths. We have now added summary statistics of GWAS performed for 22 binary and quantitative traits (Table S6-S7), an analysis of the prediction performance of polygenic scores computed using MXB and UKB GWAS (Fig. S41, Table 1, Fig. 4, Tables S8-S9, Extended Data Figures 1-10), and included polygenic scores and BMI in the complex trait models (Figs. 5-6, S53-S57). We hope, especially with these new analyses, it is more apparent that this is not only a population genetics study, but also a complex traits study, the first of such scale and breadth conducted on Mexicans living in Mexico by a team centered in Mexico. We have also gone further in the ROH analyses (Fig. 3A), as well as the analyses of birth year and ancestries (Figs S36-S38) (see response to

reviewer # 2 below), and have added archetypal analyses (Gimbernat-Mayol et al Plos Computational Biology 2022) as a novel approach to assess and visualize population structure exemplifying its continuous nature (Figs S21-23). We have now also conducted an analysis using Admixture-bayes to infer the admixture histories of the different Mesoamerican regions in Mexico (Fig. 2D). Our results provide new insights into the demographic history of Indigenous groups in Mexico. By using an ancestry-specific approach, we were able to infer admixture events that occurred prior to the colonization period and to understand the variation of admixture events within the country.

Major concerns are:

1. The models of the complex traits did not include an important measure of genomewide risk, namely the polygenic risk score (PRS). Each of the traits should be re-analyzed to include its PRS as one of the covariates to properly evaluate the contribution of genomewide risk factors currently included in the models.

We agree with the reviewer and have now re-analyzed each trait after adding its polygenic score to the model (Figs. 5-6, S53-S57).

2. There is an important key covariate (i.e. BMI) which should be included in the analysis of traits such as fasting glucose, triglycerides, HDL-cholesterol, LDL-cholesterol, systolic blood pressure and diastolic blood pressure. For these traits, age, sex and BMI are the three covariates with the largest effect sizes in most populations. The authors included age and sex, but not BMI, in their models. The analyses should be done with the inclusion of BMI. Otherwise, the presented effect sizes of the factors and model r^2 could be considered unreliable.

We agree with the reviewer and have now added BMI as a covariate to the analysis of all traits excluding height, BMI, and creatinine (Figs. 5-6, S53-S57).

3. The analysis of the genomic burden of rare deleterious mutations (and of other classes of variants) is based on array genotypes. Most arrays have a SNP ascertainment bias as well as design biases (based on the objectives of the array), and may not adequately mirror whole genome sequences (WGS). While this point is acknowledged in part in the manuscript, the current comparison with the subset of Mexicans living in Los Angeles (1000 Genomes MXL) is insufficient and is based on the assumption that the 1000 Genomes MXL are representative of most or all of the groups in the current study. The authors should discuss further how likely their findings will remain consistent if they had whole genome sequence (WGS) in the MXB participants that they studied.

We thank the reviewer for this comment which we appreciate. We have now included WGS data from MXB samples and added a new analysis. In this analysis, using 50 genomes sequenced as part of the Mexican Biobank, we correlate the rare mutation burden computed using SNPs in WGS data with rare mutation burden computed using SNPs in array data (Fig. S40). We also do

the same analysis for two other 1000 Genomes populations using either WGS SNPs or only the subset of SNPs available on the MEGA array. We observe a significant positive correlation in all cases for rare mutation burden computed using WGS or array data. Through this analysis, we help further establish that it is reasonable to compute rare mutation burden from the array data for the analysis presented in the study. The 1000G MXL are certainly not representative of most or all of the groups in the study. They show the same correlation of rare burden with native and other ancestries as the MXB samples. We think this reflects primarily the Bering strait bottleneck leading to loss of rare variants or their rise to higher frequencies leading to a lower burden of rare variants in these ancestries. In this sense, the fact that the 1000G MXL does not represent all native ancestries is not affecting our observations. These individuals do carry native ancestries, and the pattern that we show is likely common to all native ancestries due to the shared demographic history. We further show now how the MXB array data computed burden is a reasonable proxy and does show the same signal across a swath of native ancestries. We have also now further increased discussion of this important point raised by the reviewer in the main text.

Mutation burden concordance between WGS and MEGA array (MAF < %5)

Reviewer # 2

The authors based their manuscript on an interesting dataset for medical and population studies, including more than 6,000 individuals from different regions of Mexico. The manuscript covers many topics to infer the relationship between regional demography, ancestry, genetic load, environment, and health data. However, the different results are not well connected throughout the text. Although the database is quite interesting, the reported findings' novelty is unclear, all of which were briefly described and hardly discussed concerning the literature and previous results. In many moments, the text seems only to corroborate previous studies; that is, more data were used to conclude what had already been published about the Mexican population.

The sample is the key point of this manuscript. Still, unfortunately, because it is not complete genomes, it is not something new or unpublished to Mexican populations, as there are several GWAS studies using large samples already published in this population (Spear et al. 2020; Martinez-Magana et al, 2021; Feofanova et al, 2020).

As summarized by reviewer # 1, an important advance of our study is the fine-grained resolution of ancestry and admixture with the integration of archaeological and historical information. Other strengths of the study include the large sample size, representativeness of the study population, integration with archaeological and anthropological evidence, and the complex trait modeling in the context of genetic and environmental factors.

We have also now added genome-wide association studies performed for 22 binary and quantitative traits (Table S6-S7), an analysis of prediction performance of polygenic scores computed using MXB and UKB GWAS (Fig. S41, Table 1, Fig. 4, Tables S8-S9, Extended Data Figures 1-10), and included the polygenic scores and BMI in the complex trait models (Figs. 5-6, S53-S57). We hope, especially with these new analyses, it is more apparent that this is not only a population genetics study, but also a complex traits study, the first of such scale and breadth conducted on Mexicans living in Mexico by a team centered in Mexico. We have also gone further in the ROH analyses (Fig. 3A), as well as the analyses of birth year and ancestries (Figs S36-S38), and have added archetypal analyses (Gimbernat-Mayol et al Plos Computational Biology 2022) as a novel approach to assess and visualize population structure exemplifying its continuous nature (Figs. S21-23). We have now also conducted an analysis using the recent AdmixtureBayes method of Nielsen et al. (2022) to infer the admixture histories of the different Mesoamerican regions in Mexico (Fig. 2D). Our results provide new insights into the demographic history of Indigenous groups in Mexico. By using an ancestry-specific approach, we were able to infer admixture events that occurred prior to the colonization period and to understand the variation of admixture events within the country.

We have previously published about the great amount of diversity and genetic differentiation of Mexican populations (Moreno-Estrada et al 2014), pointing that finer and more comprehensive

samplings within Mexico will result in novel variation to be explored. Therefore, a nationwide genomic survey like the one we are presenting remains unpublished, regardless of consisting of SNP array or complete genomes data. The sampling coverage of our study is unprecedented as every single Mexican State is represented and thus it captures new diversity not reported before. Such diversity, linked with phenotypic and environmental data across rural and urban locations, cannot be captured by studies conducted with Mexican participants outside of Mexico. The other studies the reviewer mentioned either perform or use a GWAS in Latin Americans living in the United States (Feofanova et al 2020, Spear et al 2020). Feofanova et al (<https://www.sciencedirect.com/science/article/pii/S0002929720303232>) conduct a GWAS on 640 circulating metabolites in 3,926 Hispanic Community Health Study/Study of Latinos participants. Spear et al mainly show that native ancestries are increasing in the same cohort. Martinez-Magana et al (<https://www.nature.com/articles/s41598-021-85881-4>) is a very specific study on psychiatric disease and substance abuse conducted on 3914 Mexicans. As shown by previous work (see, for example, Mostafavi et al elife 2020), it is important to perform GWAS across diverse environments even in single ancestry groups. Our study performs GWAS across 22 binary and quantitative traits in Mexicans living primarily in rural environments (~70% of the biobank) or in urban environments in Mexico. Further, our biobank has ~1,000 individuals from across the country with primarily native ancestries with linked complex trait information, the largest genetic and complex trait dataset of indigenous individuals living in Mexico, on which we also perform and release GWAS summary statistics. Lastly, GWAS across a range of traits is only a part of our work, with other strengths described above.

We hope that by clarifying these points, and with the several new analyses in the revised manuscript, we can help the reviewer see the strengths of this particular study and the specific void that it fills.

Specifically:

1. Regarding the ancestry analyses, I have doubts about what is really new in this study. For example, Moreno-Estrada had already demonstrated in his 2015 paper, also using array data from more than 1000 mestizos and native individuals, that the native ancestry of mestizos recapitulated pre-Columbian regional origin in Mexico. The present manuscript describes the same, only including more samples from different regions.

We thank the reviewer for this comment. Compared to Moreno-Estrada et al 2014, the larger representation and diversity of native ancestries captured in this study design actually show that primarily the substructure is amongst the Mayan region and the other Mesoamerican regions (Fig. 1C and E). With respect to the rest of the Mesoamerican regions, we actually make the point that the structure is very subtly present, with mixing and migration histories actually showing a continuum of ancestry. In genetic similarity space (as represented by PCA and

UMAP), it is impossible to draw clusters reflecting pre-Columbian origin. This interpretation is only possible with the breadth of sampling across these native ancestries in the country as was done in this study, compared to the discretized sampling of Moreno-Estrada et al 2014. We have now added more Admixture analyses to further problematize the varying ancestries sources in Mexico (Figs. S11, S12), and have also added archetypal analysis (Figs. S21-23)(Gimbernat-Mayol et al Plos Computational Biology 2022) which gives an alternative view of clusters, furthering the idea of different ways to look at clusters of ancestries existing within Mexico beyond a model considering two primary European and Indigenous clusters. In this new approach, all clusters represent a real sample, or linear combination of samples (so something that could be achieved by mating), whereas with ADMIXTURE clustering can sometimes be best fit to residuals with some clusters representing allele frequencies that have never existed in any population, so they not only generate a false sense of a "pure" population, but some are not even population allele frequencies that have ever existed, just an artefact. We have now also conducted an analysis using the recent AdmixtureBayes method of Nielsen et al. (2022) to infer the admixture histories of the different Mesoamerican regions in Mexico (Fig. 2D). Our results provide new insights into the demographic history of Indigenous groups in Mexico. By using an ancestry-specific approach, we were able to infer admixture events that occurred prior to the colonization period and to understand the variation of admixture events within the country.

2. The authors describe the finding of East Asian ancestry related to the Manila galleon in Mexico. It is surprising, given that the same authors have an article entirely dedicated to the subject (Rodriguez-Rodriguez et al. 2022), quite complete (not preliminary as described in the text), and recently published.

The reviewer is correct. We have now removed a detailed description and discussion of this result, only noting it briefly.

3. The most exciting part of the ancestry analysis is related to the demographic expansions in the different periods of the pre-contact history of Mexico. However, the results are presented in a very descriptive way, without significant discussions.

We have presented these analyses at the fine-grained resolution of different Mesoamerican regions and have integrated archaeological and historical information to help in the interpretation and discussion of these results. In addition, and following the reviewer's suggestion, we have now included a novel methodology to learn more about the origin and dynamics of the sample studied. In particular, we applied the recent AdmixtureBayes method of Nielsen et al. (2022), which allowed us to infer population splits and inter-population interaction edges using the same regional groups as in Fig 2 (IBDNe), which complements our demographic inference across Mexico (Fig. 2D). We hope the new analyses and these points suffice the reviewer's concern (see details in point 4. below).

4. The methodology is quite simple; the authors could have used fine-structure and f-statistics methods to answer more profound questions about the origin and dynamics of the sample studied. The methodology used is also not new.

This paper studies demographic history and links it to genetic variation and trait-relevant variation in particular. Given this already large scope, and the new addition of GWAS results and polygenic score analyses across a range of traits (Fig. S41, Tables S6-S10, Extended Data Figures 1-10), we have now centered the novelty of our study in connecting these different analyses, to study the impact of demographic histories on trait-relevant variation, performing GWAS and polygenic score analyses in the largest cohort from across Mexico, and the joint modeling of genetic and environmental factors to study trait variation. In addition, and following the reviewer's suggestion, we have included a novel methodology to learn more about the origin and dynamics of the sample studied. In particular, we applied the recent AdmixtureBayes method of Nielsen et al. (2022), which allowed us to infer population splits and inter-population interaction edges using the same regional groups as in Fig 2 (IBDNe), which complements our demographic inference across Mexico (Fig. 2D). We hope the new analyses and these points suffice the reviewer's concern.

Using AdmixtureBayes, we generated admixture graphs that represent the demographic history of the main Mesoamerican regions of Mexico. By using an ancestry-specific approach, we were able to limit this history to genomic segments with ancestries from Central America.

In the admixture graph (Fig. 2D), we can observe a clear progression of splits among the populations from north to south with the North of Mexico splitting first, followed by the Occident of Mexico, the Central region, Oaxaca, the Gulf region, and the Maya region. Interestingly, the Gulf and Mayan regions are related, consistent with prior suggestions based on IBD¹² and both share a common ancestral source with, most recently, Oaxaca. The North of Mesoamerica is the result of an admixture event that occurred between the n6 (ancestral to all of Mexico except the north) and n4 (ancestral to central, eastern, and southern Mexico) nodes, with a 70% and 30% proportion respectively. This pattern corresponds with the geography and history of the regions.

5. The ROH analyses seem really interesting, however, there seems to be some confusion regarding the theoretical part and subsequent interpretation of data. Mainly regards the inference of time and number of ROH. In general, this section is not clear.

For example, the following sentence seems odd:

A bottleneck event or a long-term small population size will result in a large number of small ROH.

The correct would be to relate the short ROH to ancient events. Usually, bottleneck or isolation creates specific haplotypes of a population. Short ROH measuring tens of kb probably reflects homozygosity for ancient haplotypes that contribute to local LD patterns. In other words, it would be necessary to have an extreme and ancient bottleneck (like those explained by the out-

of-Africa). Or a population to have had its effective size reduced many generations ago (redundant with the bottleneck) to generate a large number of small ROHs. However, this doesn't need to occur over a long period in a small population. Also, for example, many native populations have expanded but still have many short ROHs.

We agree with the reviewer and have now edited this sentence to be clearer, in light of the reviewer's comments to:

The appearance of many small ROHs indicates coalescences occurring at a period in the more distant past. For, example, due to an ancient bottleneck or relatively small population size in the past. The appearance of large ROHs indicates a more recent bottleneck or many endogamous matings.

In the following sentence:

Consanguinity or marriage between relatives would instead result in fewer but longer...

It will be better to reframe it as:

Consanguinity or marriage between close relatives would instead result in fewer but longer...

We agree and have reframed this sentence as suggested by the reviewer.

I strongly suggest the use of different sizes of fragments in the analysis (at least three classes) and consider small ROH only those smaller than 2Mb.

We already use 3 classes (Fig. 3B, C) (1-8Mb, 8-12Mb, >12Mb). We chose these categories based on Ringbauer et al 2021 which analyzed the data in the same bins, and motivated the bins using theoretical modeling (>12Mb most likely reflects consanguinity, 8-12 Mb reflects recent population size and endogamy, and 1-8Mb reflects more ancient population size).

To the reviewer's suggestion of looking at ROH smaller than 2Mb, we already do this in the 1-8Mb range. We are reluctant in going any shorter due to concerns with accuracy of calling and interpreting ROH shorter than 1Mb (for example, the population software plink does not return ROH smaller than 1Mb for these reasons).

We apologize for the confusion that Fig. 3A showed all ROH size bins together. For that figure as well, we now add our three ROH classes of different sizes.

Still, concerning ROH, can the fact that young individuals have more Central American fragments be an effect of sample bias? How much of the population studied are young people? Is the age and region distribution homogeneous?

As suggested by the reviewer, we now show the following plots (Figs. S36-S38). We do have a sampling bias towards younger individuals in the Mexican Biobank (Fig. S36). The age distribution in rural and urban areas is not homogeneous (Fig. S37). Individuals in rural areas are significantly older than individuals in urban areas (Wilcoxon test $W = 3221193$, $p\text{-value} = 6.518e-07$), and we observe significantly higher ancestries from Central America in rural areas compared to in urban areas (Fig. S38, Wilcoxon test $W = 3972985$, $p\text{-value} < 2.2e-16$).

Therefore, if there is any sampling bias, it is in the opposite direction of the signal we observe, that is it would make it such that older individuals have more ancestries from Central America as we have sampled more older individuals from rural areas where Central American ancestries are more prevalent. We, instead, observe that younger individuals have more ancestries from Central America. This is also what was observed by Spear et al eLife 2020 in an entirely different dataset in Mexican-Americans.

6. The association between genomic data, phenotype and environment is poorly explored, without a proper discussion of the results. Furthermore, none of the associated phenotypes is new or unheard of for Mexican populations but the authors did not cite any previous articles published on the Mexican populations.

We have now revised the complex trait models significantly. We have added polygenic scores for each trait to all trait models, and BMI as a covariate to the relevant trait models (Figs. 5-6, S53-S57). We also further perform analyses in the MXB of the previously reported ABCA1 variant to aid in interpretation of Cholesterol results (Fig. S59). We have significantly edited the discussion as well. It is novel to model phenotypes, genomic data and environmental factors as we do and we now highlight that further.

Reviewer #3

In this manuscript, entitled ‘Nationwide genomic biobank in Mexico unravels demographic history and complex trait architecture from 6,057 individuals’, Sohail and colleagues reports genome-wide data from 6,057 individuals from Mexico. Through the genotyping of 1.8 million markers in all individuals, who come from 32 states in Mexico, they explore variation across Mexicans in ancestry proportions, IBD segments, and runs of homozygosity. Finally, they correlate ancestry and ROH with trait variation, focusing on some interesting traits such as height, BMI, and some other metabolism-related traits.

The manuscript is very well written, and I applaud all the aspect on diversity inclusion. Not only the genomics literature is strongly biased towards individuals of European ancestry but, in the case of Mexico, ancestry studies are particularly relevant because different ancestries are present, at varying extents, in the same populations. For me, this study presents a great resource for future diversity and genomic studies.

Having said that, I think that the study, in its current state, remains too preliminary and descriptive. Besides the larger sample size reported (which is great), most observations have already been published by the same authors (Rodriguez-Rodriguez et al. *Philos Trans R Soc Lond B Biol Sci* 2022; Ojeda-Granados et al. *Mol Biol Evol* 2022 <https://academic.oup.com/mbe/article/39/1/msab290/6379730>; Moreno-Estrada et al. *Science* 2014).

We thank the reviewer for the positive comments, and have now added significant new analyses and edited the text and discussion. We have now added summary statistics of GWAS performed for 22 binary and quantitative traits (Table S6-S7), an analysis of prediction performance of polygenic scores computed using MXB or UKB GWAS (Fig. S41, Table 1, Fig. 4, Tables S8-S9, Extended Data Figures 1-10). Both GWAS and Polygenic score analyses are novel additions

with respect of previous publications on Mexican populations and thus provide new insights on the genetics of complex traits. We have also included the polygenic scores for each trait and BMI as a covariate in the complex trait models (Figs. 5-6, S53-S57). Additionally, we have gone further in the ROH analyses, as well as the analyses of birth year and ancestries, and have added archetypal analyses (Gimbernat-Mayol et al Plos Computational Biology 2022) as a novel approach to assess and visualize population structure exemplifying its continuous nature (Figs. S21-23). We have now also conducted an analysis using the recent AdmixtureBayes method of Nielsen et al. (2022) to infer the admixture histories of the different Mesoamerican regions in Mexico. Our results provide new insights into the demographic history of Indigenous groups in Mexico. By using an ancestry-specific approach, we were able to infer admixture events that occurred prior to the colonization period and to understand the variation of admixture events within the country.

This is the reason why I think the authors should consider the following major points, before proceeding further:

- Besides the descriptive analyses of PCA and admixture, I think a proper demographic inference would be a great asset for the paper, and I think the authors should perform proper demographic modeling with the data. In its current state, it remains just descriptive, and one learns very little about the demographic history of the different ancestries observed in Mexico.

This paper studies demographic history and links it to genetic variation and trait-relevant variation in particular. Given this already large scope, and the new addition of GWAS results (Tables S6-S7) and polygenic score analyses across a range of traits (Fig. S41, Extended Data Figures 1-10, Table 1, Fig. 4, Tables S8-S9, Figs. 5-6, S53-S57), we have now centered the novelty of our study in connecting these different analyses, to study the impact of demographic histories on trait-relevant variation, performing GWAS and polygenic score analyses in the largest cohort from across Mexico, and the joint modeling of genetic and environmental factors to study trait variation.

Further, array data does not lend itself to a proper demographic inference (a subject of another project using WGS of a subset of the individuals presented in this paper). However, and following the reviewer's suggestion, we have included a novel methodology that is suitable for SNP array data to perform demographic modeling with the available sample set. In particular, we applied the recent AdmixtureBayes method of Nielsen et al. (2022), which allowed us to infer population splits and inter-population interaction edges using the same regional ancestry groups as in Figure 2 (IBDNe), which complements our demographic inference across Mexico (Fig. 2D).

In brief, Figure 2 presents analyses of population size history (using ancestry specific IBD tracts) at the fine-grained resolution of different Mesoamerican regions and have integrated archaeological and historical information to help in the interpretation and discussion of these results. In light of the reviewer's comment, we have now further strengthened the discussion of this section. We have also analyzed demographic histories in different mesoamerican regions using the new HapNe approach.

Further, using AdmixtureBayes, we generated admixture graphs that represent the demographic history of the main Mesoamerican regions of Mexico. By using an ancestry-specific approach, we were able to limit this history to genomic segments with ancestries from Central America.

In the admixture graph (Fig. 2D), we can observe a clear progression of splits among the populations from north to south with the North of Mexico splitting first, followed by the Occident of Mexico, the Central region, Oaxaca, the Gulf region, and the Maya region. Interestingly, the Gulf and Mayan regions are related, consistent with prior suggestions based on IBD¹² and both share a common ancestral source with, most recently, Oaxaca. The North of Mesoamerica is the result of an admixture event that occurred between the n6 (ancestral to all of Mexico except the north) and n4 (ancestral to central, eastern, and southern Mexico) nodes, with a 70% and 30% proportion respectively. This pattern corresponds with the geography and history of the regions.

We hope the new analyses and these points suffice the reviewer's concern.

- The ancestry-specific analyses for East Asian and South Asian ancestries in Mexicans deserves further attention and supplementary analyses. Given that they are present in quite low frequencies in the populations, the authors should verify more formally their geographic origins and timing. For example, 1) how sensitive is the local ancestry inference to detect very low percentage ancestry tracts, e.g., here South Asian and East Asian tracts (moreover, South Asian is not super distant from European, and East Asian has also shared ancestry with American)? 2) given the low percentage, after masking other ancestries, the remaining South Asian/East Asian ancestry-specific SNPs must be very few and with lots of missing data, how confident of the analyses the authors are when dealing with such high missing data? To deal with both 1) and 2), I think the easiest way would be to run local ancestry inference on any published data that is known to have low percentage of East Asian/South Asian admixed ancestries; otherwise, simulation data would be helpful.

We appreciate the reviewer's concern, which was also raised by reviewer #2. Given the addition of significant new analyses of GWAS and polygenic scores, and the detailed presentation of the Asian ancestry related results in Rodriguez-Rodriguez et al. 2022, we have removed the detailed results and discussion of this observation from our study, only noting it briefly in the revision.

- The authors say that the identified correlation between rare variants and ancestries is not affected by ascertainment bias but total mutation - ancestries correlation is. To me this

contradicts the intuition of WGS vs. array. This should be supplemented with more formal analyses to show that the genotyping array used is free of ascertainment bias (see Fig S50).

We thank the reviewer for this comment which we appreciate.

We showed that the rare variant – ancestry correlation is unaffected by ascertainment by repeating it in 1000G MXL whole genome sequences and getting the same result. Similarly, for the total mutation- ancestry correlation, the correlation disappeared in whole genome data.

To further address the reviewer’s concern, we have now added a new analysis. In this analysis, we correlate the rare mutation burden using all SNPs in WGS or only SNPs in array data for 50 genomes sequenced as part of the Mexican Biobank (Fig S40). These 50 genomes are a subset of the SNP array genotyped MXB samples and have been selected to be representative of the nationwide geographic coverage within Mexico. We also do the same analysis for two other 1000 Genomes populations. Through this analysis, we help further establish that it is reasonable to rare compute burden from the array data for the analysis presented in the study. The 1000G MXL are certainly not representative of most or all of the groups in the study. They show the same correlation of rare burden with native and other ancestries as the MXB samples. We think this reflects primarily the Bering strait bottleneck leading to loss of rare variants or their rise to higher frequencies leading to a lower burden of rare variants in ancestries native to Americas. In this sense, it is not important that the 1000G MXL does not represent all native ancestries. These individuals do carry native ancestries, and the pattern that we show is likely common to all native ancestries due to that shared demographic history. We further now show how the MXB array data computed rare burden is a reasonable proxy and does show the same signal across a swath of native ancestries. We have also now further increased discussion of this important point raised by the reviewer in the main text.

- Importantly, although they correlate ancestry with variation in phenotypic traits, they do not report any GWAS data/statistics. I don’t understand why they haven’t gone further in these analyses. Given the sample size, I think, particularly for Nature, it is compulsory that they perform GWAS for each of the traits reported.

We agree with the reviewer on this observation and following their suggestion we have now added analyses of GWAS performed for 22 binary and quantitative traits (Tables S6-S7) (releasing these summary statistics publicly with the manuscript). We have also added an analysis of prediction performance of polygenic scores computed using MXB or UKB GWAS (Fig. S41, Table 1, Fig. 4, Tables S8-S9, Extended Data Figures 1-10), and included the polygenic scores for each trait and BMI as a covariate in the complex trait models (Figs. 5-6, S53-S57).

- The regression analyses are presented in a very descriptive way; they should be interpreted. We have now updated the trait models as described above by including polygenic scores for each trait and BMI as a covariate (Figs. 5-6, S53-S57). We further perform analyses in the MXB of

the previously reported ABCA1 variant to aid in interpretation of Cholesterol results (Fig. S59). We have significantly edited the text to provide further interpretation.

- The method to define IBD (IBDNe) has been recently criticized (e.g., <https://doi.org/10.1101/2022.08.03.501074>), so I think the authors should cross-check the validity of their results with an independent method.

HapNe is not ancestry-specific and would produce “biased estimates for data sets including a history of strong recent admixture” as the authors note in the study. Nevertheless, taking the reviewer’s suggestion into account, we repeated the analysis with this method in the subset of individuals with more than 98% ancestries from Central America to avoid admixture as much as possible, allowing us to skip the ancestry-specific step that is not compatible with HapNe. We performed this analysis in different Mesoamerican regions and compared to results from IBDNe (non ancestry specific). This resulted in HapNe giving no demographic inference at all for some Mesoamerican regions, and giving results that are overlapping with IBDNe in some other cases, while underestimating founder effects. This indicates that IBDNe better supports and recapitulates well known historical demographics, so the inferred effective population sizes are likely more accurate. We present the comparison here for the reviewer (time in generations) but opted to leave it out of the manuscript due to the observations described above. The other strength of the IBNe approach is the ability to implement an ancestry specific version of the analysis, which allowed us to use the full sample size of the MXB and overcome the limitations of the method associated with lower sample sizes. As a result, we are confident that the inferred demography is informative of the regional dynamics of different pre-Hispanic ancestries within Mexico, which is a novel result.

Minor (but important)

The link to their custom scripts/approaches

(https://github.com/msohail88/MXB_popstruct_complextraits) is not working.

The project data is now available at the European Genome-phenome Archive (EGA) through EGA accession number for study: EGAS00001005797 (Datasets: EGAD00010002361

Mexican_Biobank_Genotypes). Such link will be live upon publication. GWAS summary statistics generated as part of this study are also available at:

<https://doi.org/10.5281/zenodo.7420254>.

Reviewer Reports on the First Revision:

Referees' comments:

Referee #1 (Remarks to the Author):

The revised manuscript retains the strengths of the previous version and has been substantially improved by the addition of (a) GWAS and polygenic prediction of multiple traits, (b) evaluation of performance of PRS in the MXB in comparison with UKBB (as an alternative source of summary statistics) and TOPMed (as an alternative source of genotypes), (c) better modeling of the genetic and non-genetic contributions to trait distribution. The new WGS data (while limited in number of samples) has been useful for a better understanding of the rare variant distribution in comparison with array genotypes.

A few minor comments:

1. The authors may wish to remove the sentence: "In this sense, it is irrelevant that the 1000G MXL does not represent all native ancestries." (page 9) because it does not add to the text and the main point (that the 1000G MXL display the same correlation of rare burden with ancestries from Central America and other ancestries as the MXB samples) has already been made in the preceding sentences.
2. All gene symbols should be italicized on page 10 (main text) and throughout the text, tables, figure and supplement.
3. Age and sex are better described as demographic factors, not "life history factors" (page 11)
4. The authors may wish to simply use educational attainment as a variable in its own right, rather than as a proxy for income. In most societies, the relationship between educational attainment and income is not a simple linear monotonic relationship (as shown by their own data in Fig S52). Each of these variables is an important epidemiologic variable in its own right and the fact that they show some correlation does not necessarily mean that one should be used as a proxy for the other.
5. The genome wide significant loci on the two Manhattan plots in Figure 4 should be labeled with the gene/locus symbols.

Referee #2 (Remarks to the Author):

I appreciate the work done by the authors to address my comments. However, several points needed to be also answered in the text. The article's main points, ancestry and association analyses, still need to be clarified and better discussed in all text. No structural novelties are presented in this new manuscript version, even though new analyses have been carried out. I emphasize the two

main unsatisfactory points again:

1. Although the authors have used a new method for ancestry analyses, the results presented are mainly descriptive. These results need to be more specific, and clarify the real novelty of the method and the new data. The authors mention that the results were contextualized with archaeological, environmental, and other data. However, it remains very imprecise and little connected with the evolutionary history of the Mexican population in all aspects. There are no new insights regarding the history and demography of Mexicans explored in the text.
2. Regarding association studies, although the authors have directed the analysis to a set of associations with complex traits, the discussion is still very incipient. The most discussed topic is related to the ABCA1 gene and cholesterol levels. Considering the richness of genetic and phenotypic data, a higher discussion of the results found would be expected, even more so because these are not unprecedented results for the Mexican population. The complex traits results are not dissected as expected by the abstract.

Referee #3 (Remarks to the Author):

In this extensively revised version of the manuscript 'Nationwide genomic biobank in Mexico unravels demographic history and complex trait architecture from 6,057 individuals', Sohail and colleagues have addressed most of my previous concerns. Specifically, they have (i) conducted and reported GWAS data on 22 traits, (ii) alleviated my concerns relating to potential ascertainment bias of SNP arrays, and (iii) improved their demographic analyses. With these new analyses, I think the paper is less descriptive and more robust, so from a purely technical standpoint, I believe, the authors have done a good job in addressing the issues I raised.

Still, I think the authors could do an extra effort to streamline the whole manuscript, to better highlight the main messages. In its current state, the manuscript is a bit dull, and the main novelties of the study are somehow lost. For example, there is some unbalance in the presentation and interpretation of the results: some parts are not interpreted at all (e.g., the results of F_{st} and X-chromosome variation), while others are presented and discussed in too much length (e.g., the different traits presented in the chapter 'Complex traits display varying roles of genetics and environment'). Overall, I think this is a great resource paper, but some extra work on the structure and length of the different sections (presentation of the results and interpretation) would improve its interest and readability.

Author Rebuttals to First Revision:

Referees' comments:

Referee #1 (Remarks to the Author):

The revised manuscript retains the strengths of the previous version and has been substantially improved by the addition of (a) GWAS and polygenic prediction of multiple traits, (b) evaluation of performance of PRS in the MXB in comparison with UKBB (as an alternative source of summary statistics) and TOPMed (as an alternative source of genotypes), (c) better modeling of the genetic and non-genetic contributions to trait distribution. The new WGS data (while limited in number of samples) has been useful for a better understanding of the rare variant distribution in comparison with array genotypes.

We thank the reviewer for continued acknowledgment of the strengths of the paper and summarizing the additions in the revision.

A few minor comments:

1. The authors may wish to remove the sentence: "In this sense, it is irrelevant that the 1000G MXL does not represent all native ancestries." (page 9) because it does not add to the text and the main point (that the 1000G MXL display the same correlation of rare burden with ancestries from Central America and other ancestries as the MXB samples) has already been made in the preceding sentences.

We agree with the reviewer and have now edited and trimmed this section.

2. All gene symbols should be italicized on page 10 (main text) and throughout the text, tables, figure and supplement.

We agree with the reviewer and have made the changes.

3. Age and sex are better described as demographic factors, not "life history factors" (page 11)

We agree with the reviewer and have made the relevant change.

4. The authors may wish to simply use educational attainment as a variable in its own right, rather than as a proxy for income. In most societies, the relationship between educational attainment and income is not a simple linear monotonic relationship (as shown by their own data in Fig S52). Each of these variables is an important epidemiologic variable in its own right and the fact that they show some correlation does not necessarily mean that one should be used as a proxy for the other.

We agree with the reviewer and have made the relevant change in the text.

5. The genome wide significant loci on the two Manhattan plots in Figure 4 should be labeled with the gene/locus symbols.

We have updated the figure accordingly.

In doing our revision, we found a bug in our polygenic score analyses relating to the build of the MXB imputed data and GWAS summary statistics (hg38) and that of the UKB GWAS summary statistics and MXB genotype data (hg19). This required re-running part of the polygenic score analyses where the build was mismatched. We have updated the manuscript and all tables/figures with the new results. As expected, after lifting over the relevant datasets for matched build, we now observe better prediction performance of MXB-GWAS-based scores in MXB genotype data than before, and we observed better performance of UKB-GWAS-based scores in MXB imputed data than before. Nevertheless, our overall conclusions remain unchanged.

We also now show our Admixture Bayes inferred tree topology using the same $K=3$ model for local ancestry as used in the rest of the analyses. We have updated the relevant results section accordingly as well.

Referee #2 (Remarks to the Author):

I appreciate the work done by the authors to address my comments. However, several points needed to be also answered in the text. The article's main points, ancestry and association analyses, still need to be clarified and better discussed in all text. No structural novelties are presented in this new manuscript version, even though new analyses have been carried out. I emphasize the two main unsatisfactory points again:

1. Although the authors have used a new method for ancestry analyses, the results presented are mainly descriptive. These results need to be more specific, and clarify the real novelty of the method and the new data. The authors mention that the results were contextualized with archaeological, environmental, and other data. However, it remains very imprecise and little connected with the evolutionary history of the Mexican population in all aspects. There are no new insights regarding the history and demography of Mexicans explored in the text.
2. Regarding association studies, although the authors have directed the analysis to a set of associations with complex traits, the discussion is still very incipient. The most discussed topic is related to the ABCA1 gene and cholesterol levels. Considering the richness of genetic and phenotypic data, a higher discussion of the results found would be expected, even more so because these are not unprecedented results for the Mexican population. The complex traits results are not dissected as expected by the abstract.

We are glad the reviewer appreciated the new analyses done to address their comments. We take points 1. And 2. further into account in this new and streamlined revision. We hope the reviewer will find the new text more streamlined and specific, with more clarification and discussion of the novel aspects of our data, methods and results.

Referee #3 (Remarks to the Author):

In this extensively revised version of the manuscript 'Nationwide genomic biobank in Mexico unravels demographic history and complex trait architecture from 6,057 individuals', Sohail and colleagues have addressed most of my previous concerns. Specifically, they have (i) conducted and reported GWAS data on 22 traits, (ii) alleviated my concerns relating to potential ascertainment bias of SNP arrays, and (iii) improved their demographic analyses. With these new analyses, I think the paper is less descriptive and more robust, so from a purely technical standpoint, I believe, the authors have done a good job in addressing the issues I raised.

We are glad that the reviewer found the revised version to have suitably addressed all their previous concerns, and found the new version to be more robust and less descriptive.

Still, I think the authors could do an extra effort to streamline the whole manuscript, to better highlight the main messages. In its current state, the manuscript is a bit dull, and the main novelties of the study are somehow lost. For example, there is some unbalance in the presentation and interpretation of the results: some parts are not interpreted at all (e.g., the results of F_{st} and X-chromosome variation), while others are presented and discussed in too much length (e.g., the different traits presented in the chapter 'Complex traits display varying roles of genetics and environment'). Overall, I think this is a great resource paper, but some extra work on the structure and length of the different sections (presentation of the results and interpretation) would improve its interest and readability.

We thank the reviewer for this important comment. We have now taken the time and opportunity to streamline the manuscript in light of the comments. We hope the reviewer will agree that the new structure and length distribution along with the revised presentation and interpretation of the results improves our manuscript's interest and readability.

Reviewer Reports on the Second Revision:

Referees' comments:

Referee #3 (Remarks to the Author):

Overall, I think the authors have done a great job in streamlining the manuscript, which now reads much better. The section 'Complex trait architectures in the MXB' could be still shortened, as there are parts that could go to the methods (but I leave this to the authors/editor discretion).